# Float8@2bits: Entropy Coding Enables Data-Free Model Compression

Patrick Putzky [* 1]   Martin Genzel [* 1]   Mattes Mollenhauer [1]   Sebastian Schulze [1]   Thomas Wollmann [1]
Stefan Dietzel [1]

## Abstract

Post-training compression is currently divided into two contrasting regimes. On the one hand, fast, data-free, and model-agnostic methods (e.g., NF4 or HQQ) offer maximum accessibility but suffer from functional collapse at extreme bit-rates below 4 bits. On the other hand, techniques leveraging calibration data or extensive recovery training achieve superior fidelity but impose high computational constraints and face uncertain robustness under data distribution shifts. We introduce EntQuant, a framework that unites the advantages of these distinct paradigms. By matching the performance of data-dependent methods with the speed and universality of data-free techniques, EntQuant enables practical utility in the extreme compression regime. Our method decouples numerical precision from storage cost via entropy coding, compressing a 70B parameter model in less than 10 minutes. We demonstrate that EntQuant does not only achieve state-of-the-art results on standard evaluation sets and models, but also retains functional performance on more complex benchmarks with instruction-tuned models, all at modest inference overhead.

## 1. Introduction

Large Language Models (LLMs) have demonstrated remarkable capabilities across a wide range of tasks (Touvron et al., 2023; Grattafiori et al., 2024). Yet users seeking the best model within their constraints face a trade-off between convenient API calls and self-hosting. The latter offers critical advantages in data sovereignty and latency but demands enormous memory, as open-weight models now exceed 400 billion parameters (Grattafiori et al., 2024; Guo et al., 2025).

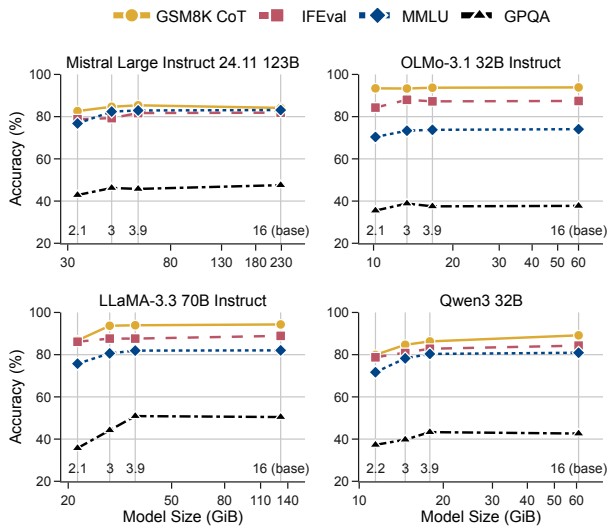

*Figure 1.* EntQuant compresses instruction-tuned models without data, performing well on several advanced benchmarks. Numbers above the size axis indicate effective bits per parameter.

Post-training quantization (PTQ) has consequently emerged as the gold standard for model weight compression (Zhu et al., 2024), reducing storage and memory-bandwidth demands while enabling faster inference (Frantar et al., 2023; Lin et al., 2024; Frantar et al., 2025).

While 8-bit quantization incurs negligible degradation (Kurtic et al., 2025; Dettmers et al., 2022; Shen et al., 2024) and 4-bit precision remains the research frontier (Kurtic et al., 2025), pushing into "extreme quantization" below 4 bits poses significant challenges. Existing methods trade accuracy against data and compute, which Nagel et al. (2019) organize into four levels. The most demanding rely on recovery training or quantization-aware training (Level 3–4) (Egiazarian et al., 2024; Tseng et al., 2024); this is problematic for specialized instruction-tuned or reasoning models, whose training data is often inaccessible or legally restricted. Calibration-based PTQ (Level 2) (Lin et al., 2024; Shao et al., 2024) is cheaper but still data-dependent and less accurate. At the other extreme, purely data-free methods (Level 1) such as rounding-to-nearest (RTN) and Half-Quadratic Quantization (HQQ) (Badri & Shaji, 2023) are fast and universal, yet collapse below 4 bits.

*Equal contribution  [1]Merantix Momentum GmbH, Berlin, Germany. Correspondence to: Patrick Putzky <patrick.putzky@merantix-momentum.com>, Martin Genzel <martin.genzel@merantix-momentum.com>.

*Proceedings of the 43rd International Conference on Machine Learning*, Seoul, South Korea. PMLR 306, 2026. Copyright 2026 by the author(s).

*Table 1.* Number of unique values in LLaMA-2 7B for different quantization levels. Averaged across layers for EntQuant. 2-bit EntQuant has more unique values than 4-bit fixed bit-width.

| Method | Quantization Levels (bits) | | |
|---|---|---|---|
| | 4 | 3 | 2 |
| Fixed bit-width | 16.00 | 8.00 | 4.00 |
| EntQuant ∅ | 63.89 | 49.06 | 34.61 |

**Quantization Bit-Width Determines Compression Rate.**
A fundamental limitation of current quantization paradigms is the strict coupling between compression rate and representational precision. Achieving a specific storage gain requires a proportional reduction in the bit-width of the weights. For instance, quantizing from `Float16` to `Int4` yields a compression factor of $4\times$. However, this relationship becomes a critical bottleneck in the extreme compression regime. To achieve $8\times$ compression, the standard framework dictates the use of a 2-bit representation, forcing the model to approximate complex weight distributions using only four distinct values. This lack of expressivity explains why standard quantization methods fail to preserve performance without explicitly accounting for outliers (Kim et al., 2024) or resorting to extensive recovery training (Egiazarian et al., 2024; Tseng et al., 2024).

**Decoupling Compression Rate from Bit-Width.** Recent advancements in lossless model compression (Zhang et al., 2025; Hao et al., 2024) suggest a solution path: modern GPUs can accelerate entropy coding, leading only to modest latency penalties. This invites a pivotal question: *Can we leverage entropy coding to break the strict coupling between compression rate and representational precision?*

In this work, we propose EntQuant (Entropy Coding Meets Quantization), a PTQ framework that answers this question affirmatively. Instead of forcing weights into a strict low-bit representation (e.g., 2 bits) to achieve extreme compression, we maintain a higher-precision representation (e.g., `Float8` or `Int8`) but optimize the weight values for *entropy*. We then employ a GPU-optimized coding algorithm based on Asymmetric Numeral Systems (ANS) (Duda, 2013) to losslessly compress these low-entropy weights; see Figure 2 for a visual overview of EntQuant.

This approach effectively decouples storage cost from representational precision. By optimizing for entropy within an 8-bit format, EntQuant achieves arbitrary bit-rates per parameter, down to or even below 2 bits while running inference on high-precision kernels. Notably, the full dynamic range of the base format remains available during inference, enabling expressivity that would be impossible under rigid low-bit quantization (see Table 1). We emphasize that these benefits extend to practical deployment scenarios as shown in Figure 1: EntQuant maintains strong performance on instruction-tuned models across challenging benchmarks, providing evidence that entropy coding offers a path beyond the limitations of fixed bit-width quantization.

**Technical Challenges.** We identify two key challenges to successfully employ entropy coding for model compression:

*Scalable Optimization of Discrete Entropy:* Directly minimizing the entropy of a weight matrix is non-differentiable and computationally difficult. We address this by formulating a relaxed optimization objective using the $\ell_1$-norm as a differentiable proxy for entropy, enabling rapid convergence with standard gradient-based solvers.

*Real-Time Decoding:* Historically, entropy coding has been viewed as a passive storage optimization, applied offline to save disk space, rather than an active component of the inference pipeline (Han et al., 2016). Inspired by the insights from DFloat11 (Zhang et al., 2025), we integrate a parallelized ANS decoder directly into the inference pipeline of the model, decompressing weights *on-the-fly* with manageable computational overhead.

**Contributions.** Apart from addressing the aforementioned challenges, our main contributions are as follows:

1. *Calibration-Free Extreme Compression:* EntQuant enables extreme PTQ (down to effective 2-bit rates) without recovery training or calibration, making it a *Level-1* method in the above taxonomy and uniquely suitable for specialized (instruction-tuned or reasoning) models where data availability is a constraint.

2. *Decoupling Compression Rate from Bit-width:* EntQuant introduces a method that separates the compression rate from the quantization bit-width, allowing for arbitrary compression rates while maintaining the expressiveness of `Float8` or `Int8`.

3. *Simplified Outlier Handling:* Unlike methods requiring explicit outlier detection (Dettmers et al., 2022) or complex grouping schemes, EntQuant uses only channel-wise scaling, letting the entropy optimization naturally concentrate precision where needed.

4. *High-Speed Optimization:* The EntQuant compression stage is highly efficient, requiring only seconds per layer, similar to fast methods like HQQ (Badri & Shaji, 2023), making it practical for immediate deployment.

**Data-Free Compression Is Essential in Practice.** Several practical scenarios make a fully data-free approach uniquely valuable. The most immediate is resource-constrained self-hosting, where memory is the binding constraint and calibration infrastructure is typically unavailable to end users (Zhang et al., 2025). Calibration data itself

is often out of reach: powerful instruction-tuned models such as LLaMA-3.3 Instruct (Grattafiori et al., 2024) and Mistral Large (Mistral AI, 2024) do not expose their training corpora, and regulated domains such as healthcare or finance are subject to data-protection rules (e.g., GDPR) that restrict the repurposing of sensitive data. Even when data is accessible, calibration can degrade alignment properties of safety-tuned and reasoning models in unpredictable ways (Lee et al., 2025; Wee et al., 2025; Kharinaev et al., 2025). Finally, with frontier releases now arriving on a weekly cadence, multi-hour calibration pipelines have become a recurring bottleneck that a data-free method requiring less than 10 minutes eliminates entirely.

**Software and Accessibility.** Code is available under https://github.com/merantix-momentum/entquant. All figures have been made colorblind safe using Paul Tol's Color Palette (Tol, 2021).

**Conflict of Interest Disclosure.** All authors are employed by Merantix Momentum GmbH. The models evaluated in this work are publicly available open-weight models from third parties; none were developed by the authors' organization. No other conflicts of interest are declared.

## 2. Method

In this section, we introduce EntQuant (Entropy Coding Meets Quantization), a method designed to minimize the storage footprint of Large Foundation Models while maintaining downstream performance. Operating on a byte-level quantization format (Float8 or Int8), we optimize the entropy of the quantized weights to approach a desired effective number of bits per parameter. EntQuant integrates the decoding step directly into the forward pass with low latency overhead.

### 2.1. Preliminaries

**Weight Quantization.** Model compression via weight quantization reduces the precision of model weights from a high-precision format, e.g., Float16 or BFloat16, to a lower-bit format such as Int8, Float8, or even Int4. Given a target quantization format $\gamma$ and an $M \times N$ weight matrix $\mathbf{W}$, we denote a quantizer by $\mathbf{W}_q = Q_\gamma(\mathbf{W}, \boldsymbol{\theta})$, where $\boldsymbol{\theta}$ is a set of parameters determining the map of $\mathbf{W}$ to its quantized form.

Dequantization brings the quantized matrix $\mathbf{W}_q$ back to a high-precision format (e.g., for inference). This yields memory savings because only the lower-precision matrix $\mathbf{W}_q$ and parameters $\boldsymbol{\theta}$ need to be stored. We denote the dequantizer by $\hat{\mathbf{W}} = Q_\gamma^\dagger(\mathbf{W}_q, \boldsymbol{\theta})$. Thus, $Q$ and $Q^\dagger$ act as lossy encoders and decoders, respectively. Under such a

quantization, a linear layer takes the form

$$\mathbf{y} = \mathbf{x}\hat{\mathbf{W}}^\top + \mathbf{b},$$

where $\mathbf{x}$ and $\mathbf{y}$ are layer inputs and outputs, and $\mathbf{b}$ is a bias term. In practice, it is often unnecessary to explicitly materialize the dequantized weight matrix $\hat{\mathbf{W}}$. Instead, computation can be made more efficient by exploiting the reduced bit-width through sophisticated fused GEMM kernels like Marlin (Frantar et al., 2025). Note that, for brevity, we focus on static weight quantization here although this viewpoint is compatible with dynamic activation quantization as well.

In this work, we consider symmetric weight quantization (Wu et al., 2020) for both Float8 and Int8 data types. The quantized weight is derived by

$$Q_\gamma : \mathbf{W}_q = \text{clamp}\left(\left\lfloor\frac{\mathbf{W}}{s}\right\rceil, -Q_{\max}, Q_{\max}\right),$$

where $\lfloor\cdot\rceil$ denotes rounding to the nearest representable value in data type $\gamma$ and $s$ is a scalar that adjusts weights to a desired range. A common approach is the AbsMax Algorithm (Dettmers et al., 2022), where $s$ is chosen as

$$s = \frac{\max(|\mathbf{W}|)}{Q_{\max}}, \tag{1}$$

to maximize the range of values used under a given quantization. In the following, we treat $s$ as an tunable parameter. The dequantizer takes the form $Q_\gamma^\dagger : \hat{\mathbf{W}} = s\mathbf{W}_q$, recovering an estimate $\hat{\mathbf{W}}$ of the original weight matrix.

**Parameter Groups.** Instead of a global scaling parameter $s$ which is used for the entire matrix, it is common practice to rescale groups of parameters $\mathbf{W}^{(g)}$ separately (Frantar et al., 2023; Lin et al., 2024; Shao et al., 2024; Badri & Shaji, 2023). The set of all scale parameters $s^{(g)}$ is denoted by $S$. Generally, smaller group sizes yield better approximations of the original matrix but incur additional storage overhead through $|S|$. Intuitively, by optimizing the scale parameters, we can align weight distributions across groups, so that the entropy of the quantized weights can be reduced.

In EntQuant, we consider channel-wise scaling, i.e., one scaling factor $s^j$ per output channel, which produces only marginal memory and inference time overhead and does not require specialized kernels.

**Entropy Coding.** Shannon's source coding theorem (Shannon, 1948) establishes that the optimal average code length for lossless compression of a sequence of i.i.d. random variables $X$ is bounded by the entropy

$$H(X) := -\sum_{x\in\mathcal{X}} p(x)\log_2 p(x).$$

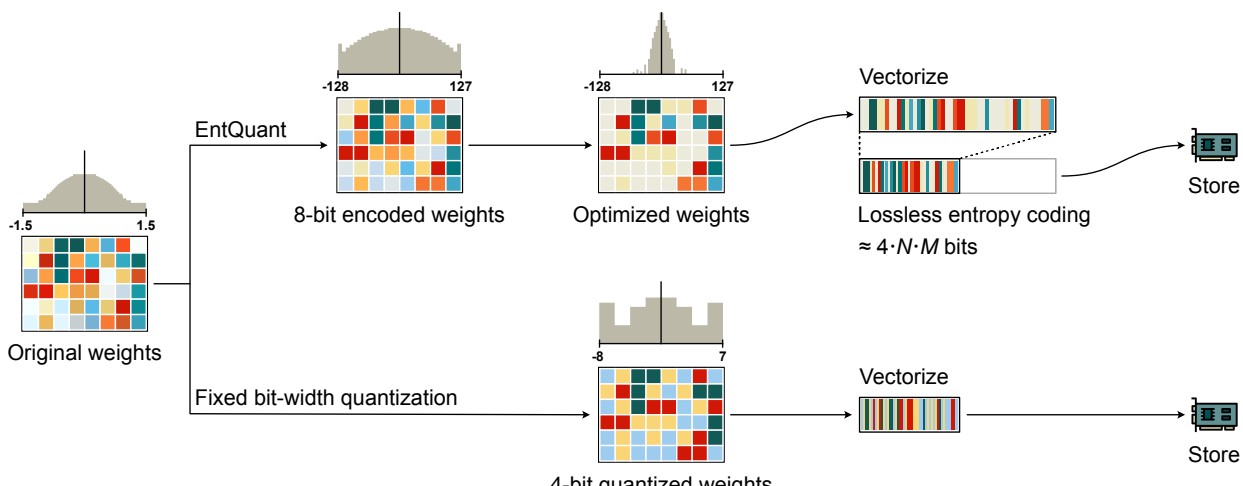

*Figure 2.* Illustration of 4-bit weight encoding with EntQuant, compared to fixed bit-width quantization. Boxes illustrate weight matrices at different representations with weight histograms above the weight matrix. Note that the number of colors and histogram bins is reduced for illustrative purposes. Weights optimized with EntQuant have more diverse parameters compared to fixed bit-width representations. With entropy coding, more common parameter values can be stored efficiently, see Table 1. Figure 3 depicts inference with EntQuant.

Several algorithms exist to approach this bound. Huffman coding (Huffman, 1952) uses variable-length prefix-free codes but is suboptimal when symbol probabilities are not negative powers of two ($2^{-k}$) or when $H(X) < 1$. Arithmetic Coding (AC) resolves these inefficiencies by encoding an entire sequence into a single rational number, offering superior compression rates (Rissanen, 1976; Pasco, 1976). However, in its basic form, AC requires computationally expensive division operations and strict serial dependencies, limiting its throughput on modern GPUs. Asymmetric Numeral Systems (ANS) (Giesen, 2014) achieves the compression ratios of AC using only faster multiplication and bit-shift operations, which makes it significantly more efficient for massive parallelization (Duda, 2013). ANS has become the basis for many modern compression algorithms, such as Zstandard (Collet & Kucherawy, 2021), and is available on NVIDIA GPUs via nvCOMP (NVIDIA Corporation, 2025) and DietGPU (Johnson, 2022), and on AMD GPUs via hipANS (Yang et al., 2024). The results of the present work are based on the nvCOMP package.

## 2.2. EntQuant: Entropy Coding Meets Quantization

The primary goal of EntQuant is to minimize the entropy of (mildly) quantized weights $\mathbf{W}_q$ to achieve stronger compression. Practical entropy coding implementations rely on the empirical distribution $\hat{p}$ under an i.i.d. assumption. For the parameters in $\mathbf{W}_q$, we assume the factorization

$$\hat{p}(\mathbf{W}_q) = \prod_{i,j} \hat{p}(\mathbf{W}_q^{(i,j)}),$$

where $\hat{p}(x) \coloneqq \frac{1}{MN} \sum_{k=1}^{MN} \delta_x(x_k)$ represents the frequency of a value $x$ in $\mathbf{W}_q$. Consequently, the empirical entropy

(expected bits per parameter) is given by

$$\hat{H}(\mathbf{W}_q) = -\frac{1}{MN} \sum_{i,j} \log_2 \hat{p}(\mathbf{W}_q^{(i,j)}). \qquad (2)$$

Ideally, we would optimize $\mathbf{W}_q$ by minimizing (2) subject to an $\epsilon$-constraint on the reconstruction error:

$$\min_{\mathbf{W}_q} \hat{H}(\mathbf{W}_q) \quad \text{subject to} \quad d(\mathbf{W}, \hat{\mathbf{W}}) < \epsilon,$$

where $d(\mathbf{W}, \hat{\mathbf{W}})$ measures the error of approximating $\mathbf{W}$ by $\hat{\mathbf{W}}$. However, solving this combinatorial problem directly is computationally challenging.

We therefore relax it into a Lagrangian formulation, also referred to as rate-distortion optimization (Sullivan & Wiegand, 1998) or entropy-constrained optimization (Chou et al., 1989):

$$\min_{\mathbf{W}_q} d(\mathbf{W}, \hat{\mathbf{W}}) + \lambda R(\mathbf{W}_q), \qquad (3)$$

where $R(\mathbf{W}_q)$ acts as a differentiable surrogate for $\hat{H}(\mathbf{W}_q)$ and the regularization parameter $\lambda > 0$ controls the compression rate.

As reconstruction loss, we use the relative, entry-wise $\ell_1$-loss, which is robust to outliers:

$$d(\mathbf{X}, \hat{\mathbf{X}}) \coloneqq ||\mathbf{X} - \hat{\mathbf{X}}||_1 / ||\mathbf{X}||_1,$$

and as regularizer, we choose the entry-wise $\ell_1$-norm:

$$R(\mathbf{X}) \coloneqq ||\mathbf{X}||_1.$$

We find that the $\ell_1$-norm serves as an effective and robust proxy for entropy reduction in all considered settings. A

formal max-entropy bound justifying this choice is derived in Section B.2. Additionally, $\ell_1$-regularization empirically induces a certain degree of sparsity in the resulting weights, see Figure B.1. Section B.1 further shows analytically that sparsity alone does not account for the observed entropy reductions, and provides a layer-wise breakdown of entropy and reconstruction error.

To solve (3), we initialize the weights using the AbsMax Algorithm from (1) and optimize each layer separately using L-BFGS (Liu & Nocedal, 1989) in PyTorch (Ansel et al., 2024). As discussed in Section 2.1, we tune only the scale parameters $S$, which leads to fast optimization even for large matrices. We use the straight-through estimator for $Q_\gamma$ (Bengio et al., 2013) to enable gradient computations through the quantization process. As shown in Figure A.1, the relationship between $\lambda$ and the target entropy is log-linear and largely model-independent. This strongly facilitates the choice of $\lambda$ in practice. Unless stated otherwise, we use `Float8` as the base quantization format in EntQuant; see Section 3.5 for an ablation with `Int8`.

**Weight encoding.** By solving (3), we obtain an optimized quantized weight matrix $\mathbf{W}_q$ and a set of scales $S$ for each linear layer. To store this data efficiently, we treat the weights as a stream of symbols to be compressed. First, the two-dimensional matrix $\mathbf{W}_q$ of dimensions $M \times N$ is flattened into a one-dimensional symbol sequence. Using an ANS coder, we compress this sequence into a compact bitstream $\mathbf{z}$. The final storage footprint of each layer thus consists of the compressed bitstream $\mathbf{z}$, the quantization scales $S$, and the metadata required by the ANS decoder (e.g., the symbol frequency table).

Since $|S| \ll M \cdot N$, the storage overhead of the high-precision (`BFloat16`) scales is negligible. Consequently, the effective compression ratio is primarily determined by the entropy of $\mathbf{W}_q$, which we explicitly minimized in (3). The full encoding procedure is summarized in Algorithm 1. Notably, this constitutes a *data-free* compression scheme, as only the weight matrix $\mathbf{W}$ is required as input.

---

**Algorithm 1** Weight Encoding Scheme

---

**Require:** Weights $\mathbf{W}$
**Ensure:** Bitstream $\mathbf{z}$, scales $S$, metadata $\mathcal{M}$
 1: $S^0 \leftarrow \text{AbsMax}(\mathbf{W})$ {Initialize with (1)}
 2: $S^* \leftarrow \arg\min_S d(\mathbf{W}, \hat{\mathbf{W}}) + \lambda R(\mathbf{W}_q)$ {Solve (3)}
 3: $\mathbf{W}_q \leftarrow Q_\gamma(\mathbf{W}, S^*)$
 4: $\mathbf{w} \leftarrow \text{vec}(\mathbf{W}_q)$ {View as 1D vector}
 5: $\mathbf{z} \leftarrow \text{ANS}(\mathbf{w})$ {Entropy encoding}
 6: **return** $(\mathbf{z}, S^*, \mathcal{M})$

---

**Inference-time Decoding.** In standard quantization pipelines, quantized weights are typically loaded directly into GPU memory. In EntQuant, we introduce an on-device decoding step that keeps weights in their highly efficient bitstream format $\mathbf{z}$ in VRAM, decompressing them *on-the-fly* when required for a forward pass. This inference procedure is formalized in Algorithm 2.

---

**Algorithm 2** Inference-Time Decoding Scheme

---

**Require:** Input $\mathbf{x}$, bitstream $\mathbf{z}$, scales $S$, metadata $\mathcal{M}$
**Ensure:** Output $\mathbf{y}$
 1: $\mathbf{w} \leftarrow \text{ANS}^{-1}(\mathbf{z}, \mathcal{M})$ {Entropy decoding}
 2: $\mathbf{W_q} \leftarrow \text{view}(\mathbf{w})$ {View as weight matrix}
 3: $\mathbf{y} \leftarrow \text{QMatMul}(\mathbf{W}_q, S, \mathbf{x}) + \mathbf{b}$
 4: **return** $\mathbf{y}$

---

While this decoding step introduces computational overhead compared to reading raw uncompressed weights, the process is designed to be highly efficient. GPU-based ANS decoders are highly parallelized, and for the large weight matrices of Foundation Models, they run with high hardware utilization. See Figure 2 above and Figure 3 below for a visualization of EntQuant's compression and inference pipeline, respectively, and Section A.1 for further implementation details and optimizations of Algorithms 1 and 2.

**Block-Wise Decompression Pipeline.** In practice, we jointly compress all entropy-optimized weights of a transformer block into a single bitstream. During inference, the model maintains one decompression buffer per device, sized to fit one transformer block in the base format (e.g., `Float8`). Before the forward pass of each block, nvCOMP jointly decompresses all weights into this buffer; individual layer weights are accessed via tensor views, avoiding copies (Line 2 of Algorithm 2). After the forward pass completes, the buffer is overwritten by the next block's weights. This block-wise scheme, following the design of DFloat11 (Zhang et al., 2025), yields an approximately 50% inference speed-up over naïve layer-wise decoding. GPU profiling (Figure A.2) shows dense alternation between decompression and forward kernels, indicating that both stages keep the device well-utilized.

**Memory Footprint.** Table 2 reports the inference-time memory breakdown for LLaMA-2 70B at 2.1 effective bits per parameter. The compressed weights dominate storage; all auxiliary components (scales, ANS metadata, decompression buffer) add less than 5% overhead. Peak GPU memory during inference is systematically analyzed in Figure F.3.

## 3. Experiments

With over 480 runs, we have evaluated EntQuant on 16 different open-weight LLMs, including LLaMA-1, LLaMA-2 (Touvron et al., 2023), LLaMA-3.1/3.3 Base & Instruct (Grattafiori et al., 2024), Qwen3 (Yang et al., 2025),

*Table 2.* Memory footprint of EntQuant for LLaMA-2 70B at 2.1 effective bits per parameter. KV cache assumes batch size 1 at the native 4096-token context with 16-bit activations.

| Component | Size |
|---|---|
| Compressed weights | ∼18.8 GiB |
| Scale parameters | ≪ 1% |
| ANS metadata | negligible |
| Decompression buffer | ∼0.8 GiB |
| KV cache | ∼1.25 GiB |

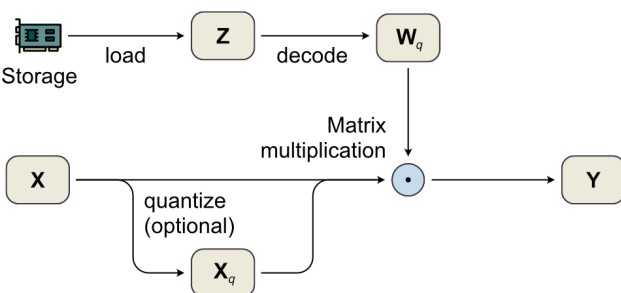

*Figure 3.* Visualization of EntQuant's inference pipeline.

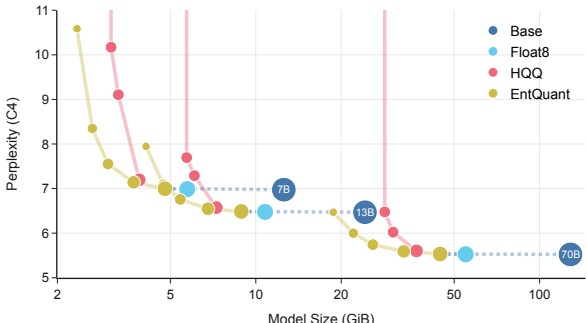

*Figure 4.* Memory-perplexity trade-off on C4 for LLaMA-2 7B, 13B, 70B. EntQuant spans a smooth Pareto front enabling fine-grained compression-performance trade-offs. Surface areas of dots are proportional to the bit-rate of each model. Float8 is entropy-encoded as well, leading to approximately 6.5 bits per parameter.

OLMo 3.1 Instruct (Team Olmo et al., 2025), and Mistral Large Instruct 24.11 (Mistral AI, 2024). Following standard practice, we report perplexity on C4 (Raffel et al., 2020) and WikiText-2 (Merity et al., 2017), as well as zero-shot accuracy on eight tasks from the EleutherAI LM Evaluation Harness (LM Eval) (Gao et al., 2023). Beyond these standard benchmarks, we assess EntQuant on instruction-tuned models using more challenging evaluations including GSM8K CoT and IFEval. Implementation details on EntQuant, baselines, and evaluations are provided in Section A.

### 3.1. EntQuant Outperforms Other Data-Free Methods

We first compare EntQuant with two popular data-free compression methods: HQQ (Badri & Shaji, 2023) and NF4 (Dettmers et al., 2023). Table 3 presents results for the compression regime where EntQuant excels, namely 2–3 bits. While EntQuant already outperforms HQQ at 3-bit precision, the gap becomes substantial in the 2-bit regime, where all baseline methods exhibit functional collapse. The Pareto front plot in Figure 4 visualizes this observation across different compression levels. Notably, at 2.1 bits, EntQuant operates below the overhead introduced by a group size of 128, which yields 2.14 bits per parameter (Chen et al., 2025, Table 11). To the best our knowledge, no other data-free method consistently performs adequately at effective 2-bit precision. The full version of Table 3 can be found in Tables C.1 to C.3, confirming that, at 4 bits, all methods perform similarly well with minor degradation. On the other hand, compressing significantly below 2 bits causes breakdowns, particularly for smaller base models.

### 3.2. EntQuant Can Compete with Calibration and Fine-Tuning Methods

We now compare EntQuant to a variety of calibration- and fine-tuning-based quantization methods. These approaches constitute a fundamentally different category of algorithms (Level 2–4 in the taxonomy of Nagel et al., 2019), typically requiring (specialized) data access and substantial compute that may not be available to end users. Table 4 shows that EntQuant performs on par with state-of-the-art fine-tuning-based methods like QuIP# (Tseng et al., 2024) and EfficientQAT (Chen et al., 2025), with only a slight gap at 2-bit precision. Full results are provided in Table D.1, including more comparison methods.

We emphasize that the above study considers only base models and relatively basic benchmarks. For instruction-tuned or reasoning models and more complex tasks, it remains unclear how calibration or recovery fine-tuning can be performed without degrading specialized capabilities. Prior work has documented that even well-tuned calibration-based quantization methods often underperform on instruction-following and hallucination detection tasks (Lee et al., 2025), and that low-bit quantization can degrade the capabilities of reasoning models (Liu et al., 2025). This stands in stark contrast to EntQuant, as we demonstrate in the next subsection.

### 3.3. EntQuant Excels on Instruction-Tuned Models

As highlighted in Figure 1, we have evaluated EntQuant in practical scenarios with models that end users would deploy. EntQuant maintains strong performance on challenging benchmarks even at 2-bit precision, while 3-bit compression incurs negligible performance drops. Our evaluation covers instruction-following (IFEval), mathematical reasoning (GSM8K CoT), scientific reasoning (GPQA), and broad knowledge (MMLU). Table E.1 presents full results

*Table 3.* Comparison of data-free compression methods on the LLaMA base model families (example: "1-7" means LLaMA-1 7B). All results are generated in-house, see also Section A.3. Best results per model and bit-rate group are in **bold**. See Tables C.1 to C.3 for full results. The full tables also report the exact memory allocations of each model, allowing for a comparison between groups sizes.

| Method | Bits | Group | C4 ↓ (Perplexity) | | | | | | | | LM Eval Avg. ↑ (Accuracy over 8 zero-shot tasks) | | | | | | | |
| --- | --- | --- | --- | --- | --- | --- | --- | --- | --- | --- | --- | --- | --- | --- | --- | --- | --- | --- |
| | | | 1-7 | 1-13 | 1-30 | 2-7 | 2-13 | 2-70 | 3.1-8 | 3.1-70 | 1-7 | 1-13 | 1-30 | 2-7 | 2-13 | 2-70 | 3.1-8 | 3.1-70 |
| Base | 16 | – | 7.08 | 6.61 | 5.98 | 6.98 | 6.47 | 5.52 | 8.43 | 5.82 | 63.5 | 66.6 | 65.3 | 64.9 | 67.9 | 72.3 | 68.9 | 73.8 |
| HQQ | 3 | 64 | 8.49 | 7.27 | 6.55 | 9.11 | 7.29 | 6.02 | 12.14 | 629.09 | 59.3 | 64.3 | 61.2 | 58.9 | 64.8 | 70.4 | 61.7 | 39.0 |
| HQQ | 3 | 128 | 9.36 | 7.60 | 6.75 | 10.17 | 7.69 | 6.48 | 14.46 | 620.89 | 58.4 | 64.1 | 61.5 | 56.8 | 63.4 | 69.5 | 58.2 | 34.1 |
| EntQuant | 3 | – | **7.52** | **6.86** | **6.18** | **7.55** | **6.76** | **5.74** | **9.74** | **6.76** | **62.4** | **65.7** | **64.4** | **63.6** | **66.7** | **71.1** | **66.0** | **72.6** |
| HQQ | 2 | 16 | 63.73 | 17.57 | 14.97 | 111.20 | 21.79 | 32.24 | 142.94 | 7.7e3 | 35.0 | 52.9 | 40.4 | 33.3 | 37.6 | 51.3 | 33.6 | 30.0 |
| HQQ | 2 | 32 | 1.4e3 | 186.58 | 60.96 | 1.2e3 | 214.22 | 323.93 | 1.9e3 | 3.0e4 | 30.6 | 32.9 | 34.0 | 29.9 | 30.3 | 31.9 | 29.9 | 29.9 |
| HQQ | 2 | 64 | 4.9e3 | 2.5e3 | 480.61 | 6.0e3 | 2.7e3 | 2.8e3 | 1.8e4 | 1.3e4 | 29.9 | 29.7 | 31.7 | 30.3 | 30.8 | 30.4 | 30.7 | 29.9 |
| EntQuant | 2.1 | – | **9.90** | **8.16** | **7.25** | **10.59** | **7.95** | **6.47** | **17.56** | **9.92** | **37.5** | **56.3** | **60.4** | **57.5** | **63.1** | **67.9** | **52.6** | **68.6** |
| EntQuant | 1.7 | – | 17.76 | 11.19 | 9.40 | 27.92 | 11.14 | 8.43 | 135.24 | 7.9e3 | 41.1 | 34.1 | 54.0 | 43.7 | 50.3 | 60.2 | 36.8 | 35.5 |

*Table 4.* EntQuant vs. calibration and fine-tuning methods on LLaMA-2 70B, comparing conceptual differences in subtable (a) and accuracy in subtable (b). Runtime specifications are taken from Frantar et al. (2023); Shao et al. (2024); Tseng et al. (2024); Chen et al. (2025), respectively. We report perplexity on C4 and WikiText-2, and avg. accuracy over five zero-shot tasks; see Table D.1 for full results.

(a)

| Method | Level | No Calibration | No Training | Compression Runtime Estimate |
| --- | --- | --- | --- | --- |
| EntQuant | 1 | ✓ | ✓ | <10min (H100)[1] |
| GPTQ | 2 | ✗ | ✓ | 2–4h (A100-80GiB) |
| OmniQuant | 2 | ✗ | ✓ | 9–16h (A100-80GiB) |
| QuIP# | 3 | ✗ | ✗[2] | ~50h (8× A100-80GiB) |
| EfficientQAT | 3 | ✓ | ✗ | ~41h (A100-80GiB) |

[1] This is a conservative estimate. Due to CPU buffering, the runtime of EntQuant also depends on the general hardware setup and utilization. Usually compression completes in significantly less than 10min and is even faster for smaller models.
[2] Recovery fine-tuning is optional but the default for QuIP#.

(b)

| Method | Bits | Group | C4 ↓ | WikiText-2 ↓ | LM Eval Avg. ↑ |
| --- | --- | --- | --- | --- | --- |
| Base | 16 | – | 5.52 | 3.32 | 72.8 |
| EntQuant | 3 | – | 5.74 | 3.62 | 71.7 (-1.6%) |
| GPTQ | 3 | 128 | 5.85 | 3.85 | 71.5 (-1.9%) |
| OmniQuant | 3 | 128 | 5.85 | 3.78 | 71.1 (-2.4%) |
| QuIP# | 3 | – | 5.67 | 3.56 | 72.1 (-0.9%) |
| EfficientQAT | 3 | 128 | 5.71 | 3.61 | 71.8 (-1.5%) |
| EntQuant | 2.1 | – | 6.47 | 4.52 | 68.6 (-5.8%) |
| GPTQ | 2 | 128 | – | – | 34.4 (-52.8%) |
| OmniQuant | 2 | 128 | 8.52 | 6.55 | 54.9 (-24.6%) |
| QuIP# | 2 | – | 6.12 | 4.16 | 70.9 (-2.6%) |
| EfficientQAT | 2 | 128 | 6.48 | 4.61 | 68.9 (-5.3%) |

across different base model sizes.

Consistent with prior observations (Lee et al., 2025), smaller models exhibit larger performance degradation under compression. These results underscore the plug-and-play nature of EntQuant: requiring no data or model-specific adaptations, our approach avoids the catastrophic failures that plague other methods when applied to realistic scenarios.

### 3.4. EntQuant Has Acceptable Inference Speed

A key difference between EntQuant and standard quantization is the on-the-fly decompression of model weights (see Figure 3 and Algorithm 2). It is therefore crucial to verify whether this overhead remains acceptable. Figure 5 shows that EntQuant is only 1.5–2× slower than the BFloat16 baseline, essentially matching NF4 inference speed while HQQ lags behind. This overhead is in line with previous implementations of entropy coding of BFloat16 weights (Zhang et al., 2025; Hao et al., 2024). Our GEMM implementation builds on the Float8 Marlin kernel (Frantar et al., 2025), denoted Float8 in Figure 5, enabling a direct analysis of the decoding overhead. For reference, CPU offloading is approximately 3× slower for prefill and 45×

slower for decoding than EntQuant.[1] Extended results for the full LLaMA-2 model family across different inference hyperparameter setups are provided in Figures F.1 to F.3. In particular, Figure F.3 confirms that peak memory gains are most significant at batch size one, and a 70B parameter model can fit into a consumer GPU (32GiB RTX 5090) at 3 bits and lower, depending on the inference load. In addition, decompression cost is independent of sequence length, since weights are decompressed once per transformer block per forward pass. Longer contexts therefore amortize the overhead. Figure 6 confirms this on LLaMA-3.1 70B at batch size 1 across prefill lengths from 512 to 8192 tokens.

### 3.5. Super Weights Help With Int8 Quantization

While the default version of EntQuant uses Float8, Int8 serves as a viable alternative base format. Figure 7 shows that Int8 achieves similar compression results to Float8 but exhibits a certain sensitivity to so-called *super weights* (Yu et al., 2024). Yu et al. (2024) identified that typically fewer than 10 particularly large outliers, occurring predominantly in early down-projection layers, cause signif-

---

[1]Tested on LLaMA-2 7B (prefill: batch size 8, seq. len. 2048 / decoding: batch size 4, context len. 512, 64 tokens generated).

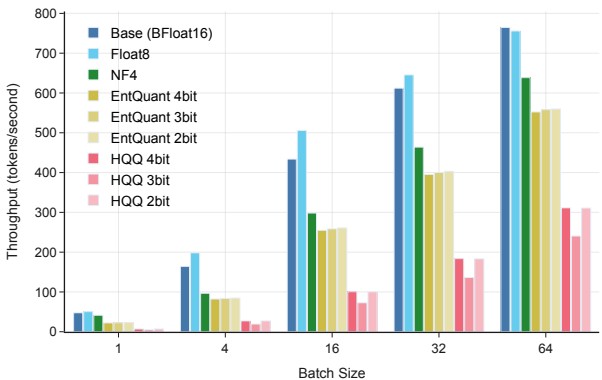

Figure 5. Inference throughput for LLaMA-2 13B in a standard prefill-decoding setting (input context length 512 tokens and 256 tokens generated). See Figures F.1 to F.3 for more results.

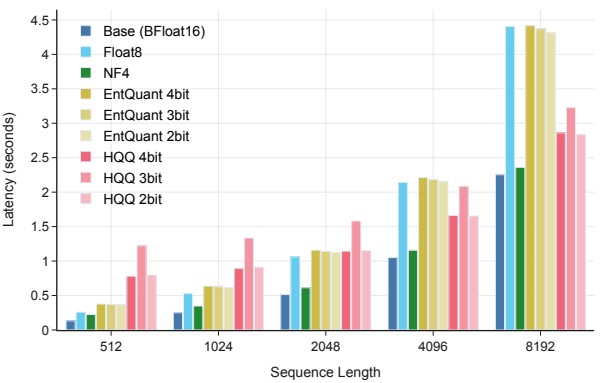

Figure 6. Inference latency for prefill on LLaMA-3.1 70B at batch size 1 across different input lengths. Since weights are decompressed once per transformer block per forward pass, the relative gap between EntQuant and the Float8 Marlin kernel on which it builds shrinks as prefill length grows, illustrating that the decompression overhead is amortized over longer contexts. Note that for pure prefill, the BFloat16 baseline is faster than Float8 and EntQuant because the Marlin kernel's advantage only manifests in decoding regimes (see Figure F.2).

icant performance drops when erased. They also propose a simple and efficient detection algorithm that requires only a single (CPU) forward pass.

Figure 7 demonstrates that precisely excluding layers containing super weights recovers expected performance for Int8. Importantly, this improvement extends beyond EntQuant: Table G.1 shows that accounting for super weights benefits NF4 and HQQ as well. Note that excluding certain layers slightly affects the overall compression ratio; for EntQuant, the reported entropy always accounts for this overhead by computing it over all linear layers. Moreover, we found that super weight handling also improves Float8 results for certain models, see Section A.2 for implementation details and model-specific thresholds.

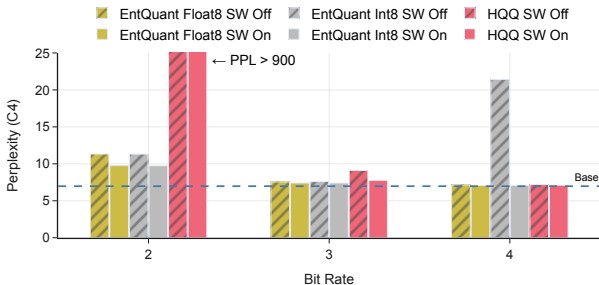

Figure 7. Comparison between EntQuant with Float8 base format (default) and Int8 for LLaMA-2 7B. Excluding layers with super weights (SW On) can substantially improve results for Int8 and modestly benefits Float8. For HQQ, perplexity still explodes in either case at 2 bits. See Table G.1 for full results.

Table 5. Weight-only quantization (W8A16) versus combined weight and activation quantization (W8A8) for the LLaMA-2 base model family (7B, 13B, 70B). We report perplexity on C4.

| Method | Bits | 2-7 | | 2-13 | | 2-70 | |
|---|---|---|---|---|---|---|---|
| | | W8A16 | W8A8 | W8A16 | W8A8 | W8A16 | W8A8 |
| Float8 | 8 | 6.99 | 7.25 | 6.48 | 6.65 | 5.53 | 6.46 |
| EntQuant | 3.9 | 7.14 | 7.28 | 6.55 | 6.68 | 5.59 | 7.10 |
| EntQuant | 3 | 7.55 | 7.76 | 6.76 | 6.88 | 5.74 | 7.35 |
| EntQuant | 2 | 11.33 | 11.59 | 8.10 | 8.23 | 6.59 | 7.90 |

### 3.6. Results for Float8 Activations (W8A16 vs. W8A8)

Weight 8-bit quantization naturally invites consideration of activation quantization. Using dynamic quantization in Quanto, we evaluate W8A8 configurations. Table 5 shows that dynamic quantization introduces a slight but acceptable performance drop, with LLaMA-2 70B exhibiting a somewhat larger gap than its smaller counterparts. Unfortunately, Quanto does not provide fused kernels for W8A8, precluding the evaluation of potential inference speedups.

## 4. Related Work

**Classical Roots in Signal Processing.** The separation of discretization (quantization) from efficient representation (entropy coding) is a foundational principle in classical information theory and signal processing (Shannon, 1948; Cover & Thomas, 2005). For decades, standards like JPEG (Wallace, 1991) have employed a pipeline where high-fidelity signals are first quantized and then losslessly compressed using Huffman or Arithmetic Coding. EntQuant can be viewed as the modern realization of this classical pipeline for Large Foundation Models: rather than quantizing DCT coefficients as in JPEG, we quantize weight matrices, replacing static codebooks with high-throughput, parallel entropy (de)coders on the GPU. In addition, we directly optimize the rate-distortion function in the context of quantization, an approach also known as *entropy-constrained quantization* (Chou et al., 1989; Gray & Neuhoff, 1998).

**Entropy Coding in Neural Networks.** In the deep learning era, *Deep Compression* (Han et al., 2016) successfully instantiated entropy coding as a useful tool for model compression after quantization of small models e.g., AlexNet (Krizhevsky et al., 2012) or VGG-16 (Simonyan & Zisserman, 2015). However, entropy coding was still an afterthought for additional storage benefits beyond quantization and pruning, without considering it for *on-the-fly* decoding. With the recent advances in entropy coding on GPUs, Zhang et al. (2025) and Hao et al. (2024) demonstrated that it can be integrated directly into the inference pipeline. Both approaches exploit the low entropy of exponent bits in `BFloat16` to achieve sizable storage reductions with manageable latency overhead. We go beyond these approaches by explicitly optimizing quantized weights to have much lower entropy.

**Fixed Bit-width in Post-Training Quantization.** Current LLM quantization methods have largely shifted to favoring fixed bit-widths, where the compression rate is strictly dictated by the underlying data type of quantized weights. Popular methods like GPTQ (Frantar et al., 2023) and AWQ (Lin et al., 2024) are highly effective at 4-bit precision with `Int4` but seem to approach a fundamental barrier at lower bit-rates (Egiazarian et al., 2024). This limitation arises from the rigid coupling of storage costs to the bit-width, which restricts the expressiveness of these methods. EntQuant breaks this limitation by turning to the classical entropy-constrained paradigm: using a high-precision data type (e.g., `Float8`) to preserve signal quality while achieving low-bit storage costs (e.g., 2.1 bits per parameter) via entropy coding (see Table 1).

**Extreme Quantization ($< 3$ bits).** To push beyond 3 bits, recent state-of-the-art PTQ methods like AQLM (Egiazarian et al., 2024) and QuIP# (Tseng et al., 2024) employ complex vector quantization combined with incoherence processing (e.g., randomized Hadamard transforms). Alternatively, quantization-aware training methods like EfficientQAT (Chen et al., 2025) achieve competitive results at 2–3 bits through end-to-end training of quantization parameters. While effective, these Level 3–4 approaches introduce significant complexity and are not data-free, making transfer to specialized instruction-tuned or reasoning models non-trivial. Moreover, achieving competitive extreme compression results typically requires the explicit handling of outliers (Dettmers et al., 2024; Kim et al., 2024), which is not case for EntQuant.

## 5. Discussion

Standard quantization couples compression rate to weight precision, so current methods either fail at extreme compression levels or require costly recovery training. EntQuant breaks this paradigm by decoupling weight precision from compression rate through entropy coding, achieving arbitrary sizes down to effective 2 bits per parameter while retaining robust model performance without calibration data or retraining. To our knowledge, EntQuant is the first method to achieve functional extreme compression in a purely data-free manner, suggesting that entropy coding can overcome the performance barriers that fixed bit-width quantization is approaching.

**Limitations.** We intentionally designed EntQuant to be the simplest version of the "Entropy Coding Meets Quantization" framework. In our pursuit of accessibility, we opted for data-free compression with simplified $\ell_1$-based regularization, foregoing complex grouping structures or custom fused operations. Many extensions are conceivable, including more sophisticated quantization schemes, advanced entropy proxies, and fused decoding kernels. While our implementation uses NVIDIA's nvCOMP library, ANS is a commoditized technology available across platforms (e.g., hipANS for AMD), ensuring broad hardware compatibility.

Our evaluation study was conducted with a budget of 5K GPU hours, in which we selected a reasonable set of benchmarks and model sizes. A more extensive evaluation on real-world tasks and larger, mixture-of-experts models is planned for future work. Since EntQuant operates on weight matrices alone, it is architecture-agnostic and applies beyond LLMs, e.g., to diffusion models (Zhang et al., 2025).

Although our implementation matches the inference speed of NF4 (Figure 5), EntQuant is currently slower than the uncompressed `BFloat16` baseline. The history of low-bit quantization is instructive here: at batch size 1, early implementations of GPTQ and QuIP were similarly slower than uncompressed inference ($\sim 0.6\times$ and $\sim 0.4\times$ TPOT, respectively), only to overtake it by a wide margin ($\sim 2.9\times$ and $\sim 3.2\times$) once dedicated fused kernels such as Marlin (Frantar et al., 2025) and E8P (Tseng et al., 2024) were introduced. EntQuant's overhead sits squarely in this pre-kernel-optimization range, and fused decompression-GEMM kernels are a natural avenue for closing the gap. Moreover, its decoding is highly parallel and compute-intensive, aligning with the continuing hardware trend where GPU compute capabilities outpace memory bandwidth growth.

**Final Remarks.** In self-hosting scenarios, memory, not inference speed, is typically the binding constraint, and users are often willing to trade latency for model quality. EntQuant directly addresses this sweet spot: by accepting a modest slowdown, it enables deployment of much larger models that would otherwise exceed memory limits. Moreover, compressing a 70B model takes less than 10 minutes without data or domain knowledge, enabling immediate adoption of newly released models.

## Acknowledgements

The authors thank Felix Möller for helpful discussions and feedback. We thank Jannis Klinkenberg and Fritz Niesel for their technical cluster support.

We kindly acknowledge funding by the European Union – NextGenerationEU – and the German Federal Ministry for Economic Affairs and Energy within the project "Souveräne KI für Europa (SOOFI)" (grant no. 13IPC040H). Computational resources were provided by the German AI Service Center WestAI and used to conduct the numerical experiments of this work.

## Impact Statement

By substantially reducing the memory required to deploy large language models, EntQuant lowers hardware costs and broadens access to powerful models for researchers and practitioners with limited computational resources. On the other hand, lowering the barrier to deployment may also facilitate the use of large models in settings where adequate safety guardrails are not in place. Future work should evaluate the interaction between extreme compression and safety-relevant model behaviors, such as toxicity and harmful-content generation.

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

# A. Implementation Details

## A.1. Implementation of EntQuant

The quantization and optimization components of EntQuant described in Section 2.2 use the Optimum Quanto package as backend. During inference with `Float8` weights (specifically `torch.float8e4m3`), we employ the Marlin kernel (Frantar et al., 2025) in the forward pass of quantized linear layers, which is natively supported by Quanto.

The regularization parameter $\lambda$ in (3) maps to a target entropy rate. While this mapping is non-linear, it is strictly monotone and robust across all linear layers and models considered; see Figure A.1 for empirical evidence. Consequently, the set of $\lambda$ hyperparameters was globally selected to match our grid of target entropies. For the learning rate of L-BFGS, we use $0.25$ when $\lambda > 30$ and $1.0$ when $\lambda \leq 30$, but in general, we found that L-BFGS is quite robust to the choice of its hyperparamters.

For `Float8`, we resolve signed zeros to eliminate unnecessary redundancy in the representation.

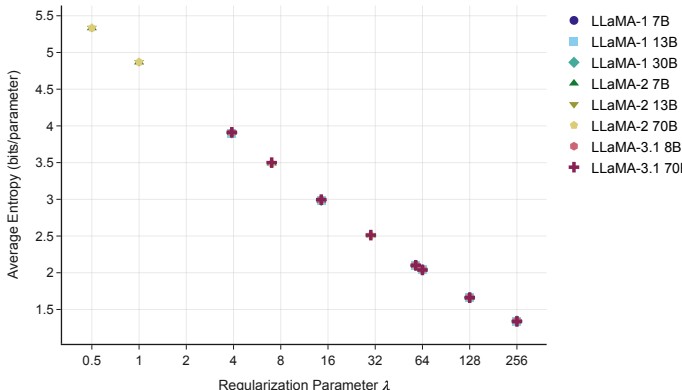

*Figure A.1.* Regularization parameter $\lambda$ in (3) vs. resulting average entropy for all EntQuant models considered in Table 3. The almost perfect clustering shows that the choices of $\lambda$ are model-independent, which drastically simplifies the hyperparameter selection process.

The block-wise decompression pipeline is described in Section 2.2. The ANS (de)compression component of EntQuant uses NVIDIA's nvCOMP package (version 5.1.0) with a chunk-size of 256KiB (NVIDIA Corporation, 2025). While each weight matrix is optimized individually via (3), Line 4 of Algorithm 1 simply concatenates all flattened attention and projection matrices into a large vector, as detailed in the main text.

GPU profiling (Figure A.2) confirms full utilization during decompression with dense alternation between decompression and forward kernels. This analysis also indicates optimization potential for multi-GPU settings. For example, upcoming blocks could be decompressed by another (idle) GPUs during the current forward pass. We leave such further optimization steps to future work.

Finally, we note that nvCOMP's implementation of ANS operates on a byte-level. Using `Float8` or `Int8` as a base format is therefore a natural choice and leads to near-optimal entropy-based encoding.

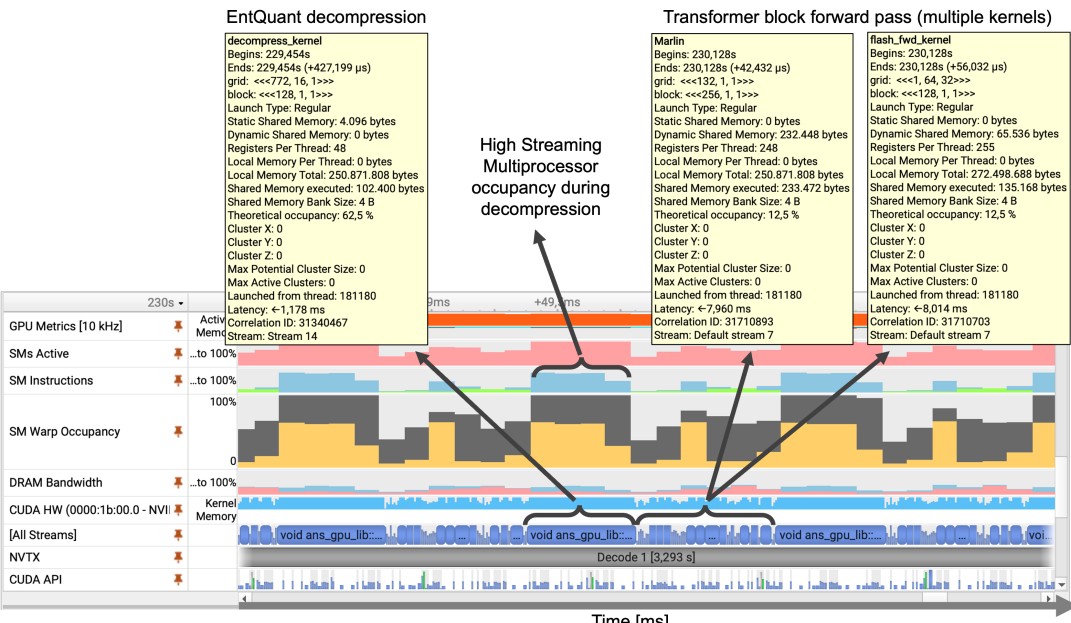

*Figure A.2.* NVIDIA Nsight™ Systems profiling of EntQuant inference (decoding) on LLaMA-2 7B showing the interleaving of ANS decompression and forward pass of each transformer block. The latter is a series of multiple forward passes of linear layers, involving kernels like Marlin (Frantar et al., 2025) and FlashAttention (Dao et al., 2022; Dao, 2024).

## A.2. Super Weights

Adopting the algorithm of Yu et al. (2024), we detect super weights via a single CPU forward pass with a dummy prompt, considering only down-projection layers as candidates. Activation thresholds were manually selected per model family:

- LLaMA-1 7B–30B: threshold $= 50$

- LLaMA-3.1 8B Base & Instruct: threshold $= 50$

- Qwen3 8B–32B: threshold $= 200$

- All other models: threshold $= \infty$ (no super weight exclusions)

Excluded down-projection layers are still quantized to 8 bits and compressed with ANS, yielding an entropy of approximately 6.5 bits per parameter.

## A.3. Baseline Methods

All experiments use PyTorch (Ansel et al., 2024) and HuggingFace's Transformers library (Wolf et al., 2020). All base models are loaded and evaluated in BFloat16 precision. Analogously to EntQuant, we use Optimum Quanto for Float8 quantization, employing the Marlin kernel (Frantar et al., 2025), see also Figure A.2. For NF4 (Dettmers et al., 2023) and HQQ (Badri & Shaji, 2023), we rely on the official Transformers integrations, which are based on the BitsAndBytes and HQQ packages, respectively. The results for the comparison methods reported in Table 4 and Table D.1 are taken from the literature and were not generated in-house, see the captions for more details.

## A.4. Evaluations tasks

**Perplexity.** We follow standard implementations for computing perplexity on C4 and WikiText-2. Context length is 2048 for LLaMA-1/2 and Qwen3, and 4096 for all other models.

**LM-Eval Benchmarks.** We use the EleutherAI LM Evaluation Harness (Gao et al., 2023) (package version 0.4.9.2) with thinking mode disabled and chat templates enabled for instruct models. Default generation settings and unmodified chat templates/system prompts are used throughout. Table A.1 below summarizes our settings for all considered evaluation tasks.

*Table A.1.* Evaluation tasks and configurations used in this work.

| Benchmark | Reference | LM-Eval Identifier | Metric | Shots |
|---|---|---|---|---|
| ARC-Easy | (Clark et al., 2018) | `arc_easy` | acc | 0 |
| ARC-Challenge | (Clark et al., 2018) | `arc_challenge` | acc | 0 |
| HellaSwag | (Zellers et al., 2019) | `hellaswag` | acc | 0 |
| WinoGrande | (Sakaguchi et al., 2021) | `winogrande` | acc | 0 |
| PIQA | (Bisk et al., 2020) | `piqa` | acc | 0 |
| BoolQ | (Clark et al., 2019) | `boolq` | acc | 0 |
| OpenbookQA | (Mihaylov et al., 2018) | `openbookqa` | acc | 0 |
| LAMBADA | (Paperno et al., 2016) | `lambada_openai` | acc | 0 |
| GSM8K CoT | (Cobbe et al., 2021) | `gsm8k_cot` | exact_match, flexible-extract | 8 |
| GPQA Main | (Rein et al., 2024) | `gpqa_main_n_shot` | acc | 5 |
| MMLU | (Hendrycks et al., 2021) | `mmlu` | acc | 5 |
| IFEval | (Zhou et al., 2023) | `ifeval` | prompt_level_strict_acc | 0 |

### A.5. Testbed Hardware

All evaluations were performed on a SLURM-based HPC cluster with NVIDIA H100 GPUs. All considered EntQuant models fit on a single H100 during evaluation. `BFloat16` base models exceeding single-GPU capacity were distributed across 2–3 GPUs using the dispatch framework of HuggingFace's accelerate package.

# B. More Conceptual Insights on EntQuant

## B.1. Low Entropy Is Linked to Sparsity

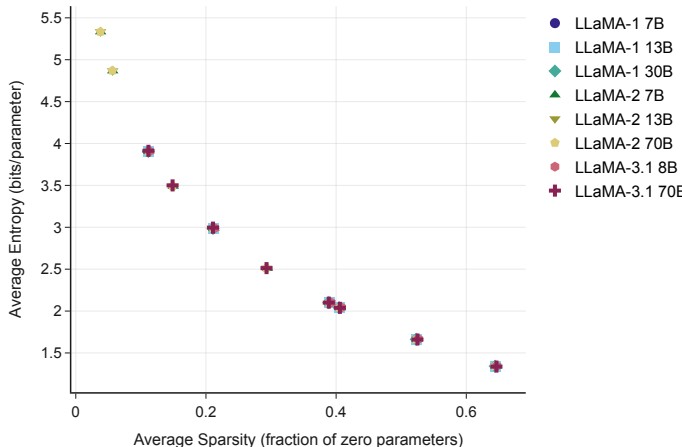

*Figure B.1.* Total sparsity over all linear layers vs. average entropy for all EntQuant models considered in Table 3. The almost perfect clustering indicates a model-independent relationship between sparsity and entropy. Hence, to a certain extent, EntQuant also operates as an (unstructured) "soft pruning" method whose compression step can also leverage non-zero weights.

**Sparsity alone does not explain EntQuant's entropy reductions.** To quantify the role of sparsity, we consider baselines where a fraction $s$ of weights are zeroed. The resulting weight distribution is a mixture of a point mass at zero (probability $s$) and a non-zero component with conditional entropy $H_{\mathrm{nz}}$, giving total entropy

$$H(s) = -s \log_2 s - (1-s) \log_2(1-s) + (1-s) \cdot H_{\mathrm{nz}},$$

where the first two terms account for the zero/non-zero indicator. We instantiate $H_{\mathrm{nz}}$ in two ways: (1) uniform over 255 Float8 levels, $H_{\mathrm{nz}} = \log_2 255 \approx 7.99$; (2) the original $\sim 6.5$-bit non-zero distribution (Figure B.1). Table B.1 shows that both baselines require sparsity $> 0.8$ to reach 2 bits, whereas EntQuant achieves 2 bits at sparsity $\approx 0.4$. This demonstrates that, under symmetric weight quantization, $\ell_1$ regularization reshapes the non-zero weight distribution toward lower entropy beyond the pure sparsity effect. Table 1 further confirms this: at 2 effective bits, EntQuant retains $\sim 35$ unique values versus only 4 for fixed 2-bit quantization, preserving the expressivity that prevents functional collapse (Table 3).

*Table B.1.* Theoretical entropy (bits per parameter) as a function of sparsity $s$ under two baselines.

|                       | $s$=0.1 | $s$=0.2 | $s$=0.3 | $s$=0.4 | $s$=0.5 | $s$=0.6 | $s$=0.7 | $s$=0.8 | $s$=0.9 |
| --------------------- | ------- | ------- | ------- | ------- | ------- | ------- | ------- | ------- | ------- |
| Uniform non-zeros     | 7.66    | 7.12    | 6.48    | 5.77    | 5.00    | 4.17    | 3.28    | 2.32    | 1.27    |
| Original distribution | 6.32    | 5.92    | 5.43    | 4.87    | 4.25    | 3.57    | 2.83    | 2.02    | 1.12    |

**Per-layer entropy and reconstruction error.** Figure B.2 shows layer-wise scatter plots for LLaMA-2 7B, relating per-layer entropy to sparsity and to weight matrix reconstruction error. We caution against correlating single-layer entropy with downstream accuracy, as layers interact non-linearly during inference, making it difficult to attribute accuracy changes to an individual layer's entropy in isolation.

## B.2. Entropy Control via a Surrogate Regularization Term

In this section, we derive a basic argument that demonstrates how the minimization of a regularization term based on the $\ell_1$-norm of a random matrix allows for control of its entropy. In accordance with the framework presented in Section 2, we focus on matrix entries with discrete distributions, which we assume to be defined over $\mathbb{Z}$. Let $X$ be a scalar random variable on $\mathbb{Z}$. The *maximum entropy principle* (Polyanskiy & Wu, 2025, Example 5.2) shows that for all $\lambda > 0$, we have

$$H(X) \leq \lambda \, \mathbb{E}[|X|] + \log Z(\lambda), \tag{B.1}$$

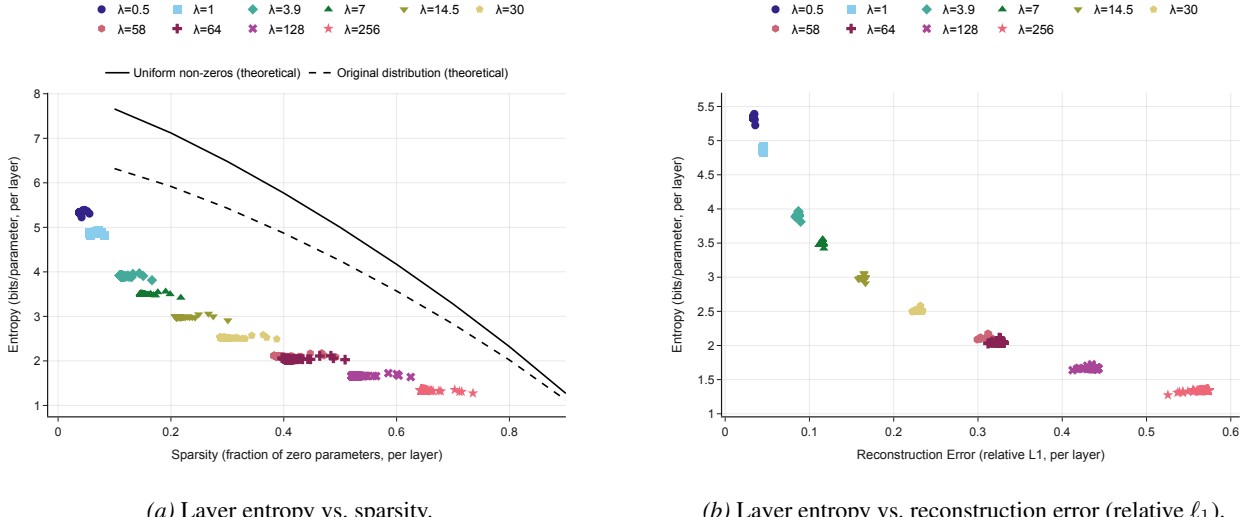

*(a)* Layer entropy vs. sparsity.       *(b)* Layer entropy vs. reconstruction error (relative $\ell_1$).

*Figure B.2.* Layer-wise scatter plots for `Float8`-quantized LLaMA-2 7B across a sweep of the entropy-regularization strength $\lambda$. Each marker is one transformer weight matrix, where color/shape encodes $\lambda$. *Left:* per-layer entropy against per-layer sparsity, with the theoretical entropy bounds for a uniform non-zero distribution (solid) and the original `Float8` value distribution (dashed) overlaid. *Right:* per-layer entropy against per-layer relative $\ell_1$ reconstruction error. Both relations are tight and monotone across the layer population, confirming that the global $\lambda$ knob induces a consistent rate–distortion trade-off at the level of individual layers.

where

$$Z(\lambda) := \sum_{n \in \mathbb{Z}} \exp(-\lambda |n|) = 2 \sum_{n=0}^{\infty} \exp(-\lambda n) - 1 = \frac{2}{1 - \exp(-\lambda)} - 1.$$

We note that this result is sharp, i.e., the bound can not be improved and is attained for a known class of Gibbs distributions. We apply this result to a random matrix $\mathbf{W}_q = (\mathbf{W}_q^{(i,j)})$ with $1 \le i \le M, 1 \le j \le N$. Using entropy subadditivity, we have

$$H(\mathbf{W}_q) \le \sum_{ij} H(\mathbf{W}_q^{(i,j)})$$

$$\le \sum_{ij} \left( \lambda \, \mathbb{E} \left[ \left| \mathbf{W}_q^{(i,j)} \right| \right] + \log Z(\lambda) \right)$$

$$= \lambda \, \mathbb{E} \left[ \| \mathbf{W}_q \|_1 \right] + MN \log Z(\lambda) \quad \text{for all } \lambda \ge 0,$$

where we used (B.1) in the second step. The nonlinear relationship between $H(\mathbf{W}_q)$ and $\| \mathbf{W}_q \|_1$ is captured in the tradeoff in $\lambda$ in the above sum, as we have $\lim_{\lambda \to \infty} MN \log Z(\lambda) = 0$. The basic intuition in the regularization context is as follows: If we assume that for some $\delta > 0$, a minimization procedure for the regularizer $\lambda R(\mathbf{W}_q) = \lambda \| \mathbf{W}_q \|_1$ over $\mathbf{W}_q$ has led to the result

$$\lambda \mathbb{E} \left[ R(\mathbf{W}_q) \right] \le \delta,$$

we have a guaranteed corresponding entropy control in terms of

$$H(\mathbf{W}_q) \le \delta + MN \log Z(\lambda).$$

This derivation also shows that the tightest worst-case control of the entropy obtained through minimization of $R(\mathbf{W}_q)$ for fixed $\lambda$ and $\delta$ depends on the ambient dimension $MN$.

# C. Full Results for Data-Free Methods

*Table C.1.* Full baseline results for the LLaMA-1 base model family (7B, 13B, 30B). We report perplexity on C4 and WikiText-2, as well as zero-shot accuracy on eight tasks from LM Eval. All results are generated in-house using the configurations described in Section A. Best results per model and bit-rate group are in **bold**. This table extends the results presented in Table 3.

| Model | Method | Bits | Group | Size (GiB) | C4 ↓ | WikiText-2 ↓ | ARC-E ↑ | ARC-C ↑ | HellaSwag ↑ | WinoGrande ↑ | PIQA ↑ | BoolQ ↑ | OBQA ↑ | LAMBADA ↑ | LM Eval Avg. ↑ |
|---|---|---|---|---|---|---|---|---|---|---|---|---|---|---|---|
| LLaMA-1 7B | Base | 16 | – | 12.6 | 7.08 | 5.68 | 74.5 | 41.6 | 56.4 | 69.4 | 78.1 | 73.6 | 40.6 | 73.9 | 63.5 |
| | NF4 | 4 | 64 | 3.9 | 7.25 | 5.82 | 74.2 | 40.9 | **55.8** | 68.2 | 77.4 | 73.1 | **42.4** | **73.4** | **63.2** |
| | HQQ | 4 | 64 | 3.9 | 7.32 | 5.91 | 72.7 | 39.4 | 53.0 | 66.4 | **77.7** | 73.3 | 38.6 | 70.7 | 61.5 |
| | HQQ | 4 | 128 | **3.7** | 7.37 | 5.96 | 73.8 | 40.6 | 55.3 | **68.5** | 77.4 | **73.6** | 41.6 | 72.4 | 62.9 |
| | EntQuant | 3.9 | – | **3.7** | **7.18** | **5.78** | **74.4** | **41.0** | 55.4 | 68.4 | 77.6 | 73.0 | 39.8 | 72.5 | 62.8 |
| | EntQuant | 3.5 | – | 3.4 | 7.27 | 5.88 | 74.5 | 41.1 | 56.6 | 69.1 | 78.6 | 75.0 | 41.2 | 72.8 | 63.8 |
| | HQQ | 3 | 64 | 3.3 | 8.49 | 6.95 | 70.2 | 36.9 | 52.5 | 66.1 | 75.8 | 69.0 | 37.2 | 66.9 | 59.3 |
| | HQQ | 3 | 128 | **3.1** | 9.36 | 7.67 | 69.8 | 36.8 | 49.3 | 64.6 | 74.5 | **71.8** | 35.8 | 64.3 | 58.4 |
| | EntQuant | 3 | – | **3.1** | **7.52** | **6.11** | **71.8** | **38.9** | **55.2** | **69.4** | **77.5** | **71.8** | **41.8** | **72.4** | **62.4** |
| | EntQuant | 2.6 | – | 2.7 | 8.10 | 6.59 | 71.7 | 37.0 | 52.8 | 64.8 | 74.7 | 72.7 | 37.8 | 68.2 | 60.0 |
| | HQQ | 2 | 16 | 3.5 | 63.73 | 68.43 | 28.9 | 21.2 | 25.8 | 49.1 | 58.5 | **61.4** | 26.4 | 8.4 | 35.0 |
| | HQQ | 2 | 32 | 2.8 | 1.4e3 | 3.2e3 | 25.3 | 20.6 | 25.4 | 48.7 | 52.7 | 45.8 | 26.4 | 0.0 | 30.6 |
| | HQQ | 2 | 64 | 2.4 | 4.9e3 | 7.7e3 | 25.0 | 21.8 | **25.9** | **49.6** | 53.1 | 38.9 | 24.8 | 0.0 | 29.9 |
| | EntQuant | 2.1 | – | **2.3** | **9.90** | **8.34** | **30.1** | **22.8** | **25.9** | 48.3 | **64.8** | 50.5 | **26.6** | **31.0** | **37.5** |
| | EntQuant | 1.7 | – | 2.1 | 17.76 | 15.67 | 49.7 | 25.8 | 34.9 | 52.9 | 65.0 | 49.9 | 27.2 | 23.6 | 41.1 |
| | EntQuant | 1.4 | – | 1.8 | 396.08 | 700.35 | 28.4 | 20.8 | 27.4 | 50.4 | 55.2 | 45.7 | 25.2 | 0.1 | 31.7 |
| LLaMA-1 13B | Base | 16 | – | 24.2 | 6.61 | 5.09 | 77.1 | 46.2 | 59.8 | 72.8 | 79.1 | 77.8 | 44.4 | 75.9 | 66.6 |
| | NF4 | 4 | 64 | 7.3 | 6.71 | 5.20 | 76.4 | 45.8 | 59.1 | 71.1 | 78.5 | **79.1** | 46.2 | **75.8** | **66.5** |
| | HQQ | 4 | 64 | 7.3 | 6.72 | 5.21 | 76.3 | 46.0 | 59.2 | 71.8 | 78.7 | 77.5 | 44.0 | 75.1 | 66.1 |
| | HQQ | 4 | 128 | 6.9 | 6.75 | 5.25 | 75.9 | **46.2** | 59.0 | 72.0 | 78.7 | 77.6 | 45.0 | 75.0 | 66.2 |
| | EntQuant | 3.9 | – | **6.8** | **6.67** | **5.15** | **76.6** | 45.4 | **59.5** | 72.2 | 79.1 | 76.7 | 44.8 | 75.6 | 66.2 |
| | EntQuant | 3.5 | – | 6.2 | 6.72 | 5.21 | 76.2 | 45.1 | 59.2 | 72.2 | 78.8 | 77.0 | 44.8 | 75.4 | 66.1 |
| | HQQ | 3 | 64 | 6.1 | 7.27 | 5.71 | 75.8 | 43.3 | 56.9 | 69.6 | 77.3 | 75.2 | 42.6 | 73.4 | 64.3 |
| | HQQ | 3 | 128 | 5.7 | 7.60 | 6.03 | 75.5 | 43.0 | 55.8 | 69.8 | 77.8 | 75.7 | 42.6 | 72.9 | 64.1 |
| | EntQuant | 3 | – | **5.5** | **6.86** | **5.36** | **76.5** | **44.7** | **59.0** | 71.5 | 78.5 | 76.9 | 44.0 | **74.2** | **65.7** |
| | EntQuant | 2.6 | – | 4.8 | 7.19 | 5.63 | 76.8 | 42.5 | 57.6 | 70.2 | 78.0 | 75.4 | 44.8 | 72.7 | 64.7 |
| | HQQ | 2 | 16 | 6.5 | 17.57 | 13.10 | **65.9** | 31.4 | 44.9 | **61.8** | 72.1 | 65.3 | **36.2** | 45.7 | 52.9 |
| | HQQ | 2 | 32 | 5.0 | 186.58 | 198.94 | 35.7 | 20.6 | 27.9 | 50.3 | 57.1 | 38.6 | 25.6 | 7.1 | 32.9 |
| | HQQ | 2 | 64 | 4.3 | 2.5e3 | 4.7e3 | 26.2 | 21.1 | 26.1 | 49.1 | 53.5 | 37.9 | 23.6 | 0.0 | 29.7 |
| | EntQuant | 2.1 | – | **4.1** | **8.16** | **6.46** | 65.7 | **35.6** | **51.8** | 60.0 | **73.6** | **70.6** | 34.2 | **59.0** | **56.3** |
| | EntQuant | 1.7 | – | 3.6 | 11.19 | 10.04 | 28.5 | 20.3 | 26.4 | 51.2 | 59.2 | 54.5 | 23.4 | 9.3 | 34.1 |
| | EntQuant | 1.4 | – | 3.1 | 69.31 | 87.73 | 35.5 | 22.3 | 32.6 | 51.1 | 57.1 | 50.7 | 24.6 | 5.4 | 34.9 |
| LLaMA-1 30B | Base | 16 | – | 60.6 | 5.98 | 4.10 | 75.9 | 45.1 | 61.3 | 66.1 | 77.9 | 81.0 | 40.2 | 74.7 | 65.3 |
| | NF4 | 4 | 64 | 17.6 | 6.08 | 4.22 | 75.5 | 44.5 | 60.8 | 67.2 | 77.8 | 81.5 | 38.8 | 74.3 | 65.1 |
| | HQQ | 4 | 64 | 17.6 | 6.08 | 4.21 | 75.5 | 45.6 | **61.3** | 67.4 | **78.3** | **82.2** | **40.6** | **74.5** | **65.7** |
| | HQQ | 4 | 128 | 16.7 | 6.09 | 4.23 | 75.6 | 44.3 | 61.2 | 66.8 | 78.0 | 81.4 | 38.8 | 73.8 | 65.0 |
| | EntQuant | 3.9 | – | **16.2** | **6.03** | **4.18** | **76.2** | **46.8** | 61.1 | **67.5** | 77.9 | 81.7 | 38.6 | 74.2 | 65.5 |
| | EntQuant | 3.5 | – | 14.7 | 6.07 | 4.22 | 75.5 | 44.0 | 61.0 | 66.9 | 78.2 | 80.3 | 40.0 | 73.7 | 65.0 |
| | HQQ | 3 | 64 | 14.6 | 6.55 | 4.82 | 71.2 | 38.5 | 56.9 | **66.2** | 76.7 | 74.3 | **37.8** | 67.8 | 61.2 |
| | HQQ | 3 | 128 | 13.7 | 6.75 | 5.06 | 72.5 | 41.3 | 57.2 | 63.5 | 75.6 | 77.4 | 35.2 | 69.3 | 61.5 |
| | EntQuant | 3 | – | **12.8** | **6.18** | **4.41** | **75.3** | **45.1** | **60.5** | 65.6 | **77.3** | **81.0** | **37.8** | **72.8** | **64.4** |
| | EntQuant | 2.6 | – | 11.1 | 6.46 | 4.80 | 74.1 | 42.2 | 59.5 | 63.8 | 77.7 | 79.4 | 36.6 | 70.3 | 63.0 |
| | HQQ | 2 | 16 | 15.7 | 14.97 | 13.96 | 43.8 | 19.7 | 36.4 | 51.2 | 63.7 | 59.4 | 29.2 | 19.6 | 40.4 |
| | HQQ | 2 | 32 | 12.0 | 60.96 | 83.88 | 32.7 | 20.6 | 28.5 | 49.6 | 56.7 | 50.4 | 27.2 | 6.5 | 34.0 |
| | HQQ | 2 | 64 | 10.1 | 480.61 | 790.32 | 27.6 | 21.8 | 26.1 | 48.5 | 54.0 | 48.9 | 26.0 | 0.5 | 31.7 |
| | EntQuant | 2.1 | – | **9.3** | **7.25** | **5.52** | **70.1** | **36.4** | **56.3** | **65.0** | **74.8** | **75.4** | **38.0** | **67.3** | **60.4** |
| | EntQuant | 1.7 | – | 7.9 | 9.40 | 7.71 | 60.7 | 30.2 | 51.5 | 61.2 | 70.8 | 68.3 | 33.2 | 56.3 | 54.0 |
| | EntQuant | 1.4 | – | 6.7 | 78.66 | 36.24 | 40.2 | 24.5 | 35.8 | 52.2 | 59.9 | 57.7 | 27.8 | 13.1 | 38.9 |

*Table C.2.* Full baseline results for the LLaMA-2 base model family (7B, 13B, 70B). We report perplexity on C4 and WikiText-2, as well as zero-shot accuracy on eight tasks from LM Eval. The 6.5-bit rate reported for Float8 results from a lossless compression step as in EntQuant. All results are generated in-house using the configurations described in Section A. Best results per model and bit-rate group are in **bold**. This table extends the results presented in Table 3.

| Model | Method | Bits | Group | Size (GiB) | C4 ↓ | WikiText-2 ↓ | ARC-E ↑ | ARC-C ↑ | HellaSwag ↑ | WinoGrande ↑ | PIQA ↑ | BoolQ ↑ | OBQA ↑ | LAMBADA ↑ | LM Eval Avg. ↑ |
|---|---|---|---|---|---|---|---|---|---|---|---|---|---|---|---|
| LLaMA-2 7B | Base | 16 | – | 12.6 | 6.98 | 5.47 | 75.5 | 42.5 | 57.1 | 69.7 | 77.7 | 79.2 | 44.4 | 73.4 | 64.9 |
| | Float8 | 6.5 | – | 5.7 | 6.99 | 5.48 | 75.8 | 43.1 | 57.2 | 69.0 | 78.1 | 79.1 | 44.4 | 73.5 | 65.0 |
| | EntQuant | 5.3 | – | 4.8 | 7.00 | 5.50 | 75.5 | 42.9 | 57.2 | 69.9 | 77.7 | 79.3 | 44.6 | 72.8 | 65.0 |
| | EntQuant | 4.9 | – | 4.4 | 7.04 | 5.54 | 76.2 | 42.2 | 57.1 | 69.8 | 78.0 | 79.1 | 43.8 | 73.4 | 64.9 |
| | NF4 | 4 | 64 | 3.9 | 7.16 | 5.65 | 75.5 | **43.2** | 56.8 | 69.9 | 77.7 | 78.8 | 44.4 | 72.5 | **64.9** |
| | HQQ | 4 | 64 | 3.9 | 7.20 | 5.68 | **75.8** | 42.1 | 56.8 | **70.1** | **78.0** | **80.0** | 44.0 | 72.3 | **64.9** |
| | HQQ | 4 | 128 | **3.7** | 7.24 | 5.74 | 75.6 | 42.9 | 56.7 | 68.8 | **78.0** | 77.1 | 43.4 | **72.9** | 64.4 |
| | EntQuant | 3.9 | – | **3.7** | **7.14** | **5.62** | 75.0 | 42.7 | **57.2** | 68.3 | **78.0** | 78.9 | **45.2** | 72.5 | 64.7 |
| | EntQuant | 3.5 | – | 3.4 | 7.26 | 5.73 | 76.1 | 42.5 | 56.4 | 69.1 | 77.5 | 78.4 | 45.2 | 72.6 | 64.7 |
| | HQQ | 3 | 64 | 3.3 | 9.11 | 7.05 | 68.5 | 37.3 | 51.0 | 65.6 | 75.1 | 68.8 | 39.6 | 65.8 | 58.9 |
| | HQQ | 3 | 128 | 3.1 | 10.17 | 7.99 | 67.2 | 33.8 | 47.4 | 65.5 | 73.3 | 67.1 | 38.0 | 61.8 | 56.8 |
| | EntQuant | 3 | – | **3.0** | **7.55** | **5.95** | **74.6** | **42.9** | **56.3** | **68.4** | **77.7** | **75.1** | **42.0** | **71.4** | **63.6** |
| | EntQuant | 2.5 | – | 2.7 | 8.35 | 6.57 | 74.5 | 40.6 | 55.0 | 68.0 | 76.6 | 71.6 | 41.6 | 68.1 | 62.0 |
| | HQQ | 2 | 16 | 3.5 | 111.20 | 115.73 | 31.0 | 19.6 | 27.3 | 49.9 | 55.5 | 52.9 | 26.0 | 3.9 | 33.3 |
| | HQQ | 2 | 32 | 2.8 | 1.2e3 | 1.6e3 | 26.8 | 21.7 | 26.0 | 49.5 | 54.1 | 37.8 | 23.0 | 0.0 | 29.9 |
| | HQQ | 2 | 64 | 2.4 | 6.0e3 | 7.5e3 | 25.3 | 21.0 | 26.1 | 50.4 | 52.8 | 41.7 | 25.2 | 0.0 | 30.3 |
| | EntQuant | 2.1 | – | **2.3** | **10.59** | **8.29** | **68.9** | **36.2** | **51.6** | **65.7** | **73.7** | **67.9** | **42.2** | **54.0** | **57.5** |
| | EntQuant | 1.7 | – | 2.0 | 27.92 | 22.48 | 52.7 | 26.7 | 43.3 | 57.4 | 67.1 | 42.2 | 32.2 | 28.0 | 43.7 |
| | EntQuant | 1.3 | – | 1.8 | 2.4e3 | 2.6e3 | 27.4 | 21.2 | 26.7 | 49.3 | 54.4 | 40.9 | 26.4 | 0.2 | 30.8 |
| LLaMA-2 13B | Base | 16 | – | 24.2 | 6.47 | 4.88 | 78.9 | 47.2 | 60.2 | 72.2 | 79.7 | 82.6 | 45.6 | 76.5 | 67.9 |
| | Float8 | 6.5 | – | 10.8 | 6.48 | 4.89 | 78.9 | 47.4 | 60.3 | 72.3 | 79.3 | 82.0 | 45.2 | 76.3 | 67.7 |
| | EntQuant | 5.3 | – | 8.9 | 6.49 | 4.90 | 78.9 | 47.4 | 60.1 | 72.5 | 79.3 | 82.2 | 46.0 | 76.4 | 67.8 |
| | EntQuant | 4.9 | – | 8.2 | 6.50 | 4.91 | 79.0 | 46.9 | 60.1 | 72.4 | 78.9 | 82.4 | 45.6 | 76.5 | 67.7 |
| | NF4 | 4 | 64 | 7.3 | 6.57 | 4.98 | **79.3** | 46.7 | 59.8 | **72.4** | 78.7 | 82.6 | 45.0 | 76.2 | **67.6** |
| | HQQ | 4 | 64 | 7.3 | 6.57 | 4.98 | 79.0 | 46.7 | **59.9** | 72.3 | 78.3 | 81.3 | 44.6 | 75.8 | 67.2 |
| | HQQ | 4 | 128 | 6.9 | 6.60 | 5.00 | 78.2 | 45.2 | 59.6 | 71.8 | 78.6 | 81.8 | **46.0** | **76.8** | 67.2 |
| | EntQuant | 3.9 | – | **6.8** | **6.55** | **4.97** | 78.1 | **46.8** | **59.9** | 72.3 | **78.7** | 82.5 | 45.0 | 76.5 | 67.5 |
| | EntQuant | 3.5 | – | 6.2 | 6.61 | 5.01 | 78.3 | 46.3 | 59.8 | 72.5 | 78.8 | 81.8 | 46.4 | 75.8 | 67.5 |
| | HQQ | 3 | 64 | 6.1 | 7.29 | 5.60 | 76.0 | 41.9 | 57.3 | 69.8 | 77.2 | 80.5 | 42.8 | 72.9 | 64.8 |
| | HQQ | 3 | 128 | 5.7 | 7.69 | 5.87 | 74.3 | 41.5 | 56.4 | 67.3 | 76.6 | 78.0 | 41.2 | 72.1 | 63.4 |
| | EntQuant | 3 | – | **5.4** | **6.76** | **5.15** | **77.3** | **44.3** | **59.4** | **72.1** | **78.5** | **82.0** | **45.2** | **74.6** | **66.7** |
| | EntQuant | 2.5 | – | 4.7 | 7.10 | 5.44 | 77.0 | 43.9 | 57.7 | 70.7 | 78.3 | 80.2 | 43.6 | 73.3 | 65.6 |
| | HQQ | 2 | 16 | 6.5 | 21.79 | 17.73 | 42.4 | 21.8 | 33.3 | 50.0 | 60.1 | 56.5 | 24.6 | 12.4 | 37.6 |
| | HQQ | 2 | 32 | 5.0 | 214.22 | 302.92 | 27.3 | 20.1 | 26.8 | 49.7 | 54.7 | 37.8 | 25.0 | 1.2 | 30.3 |
| | HQQ | 2 | 64 | 4.3 | 2.7e3 | 3.8e3 | 26.6 | 19.8 | 26.0 | 49.2 | 52.2 | 46.8 | 25.6 | 0.0 | 30.8 |
| | EntQuant | 2.1 | – | **4.1** | **7.95** | **6.24** | **74.5** | **41.4** | **55.6** | **69.6** | **76.9** | **75.2** | **41.4** | **69.8** | **63.1** |
| | EntQuant | 1.7 | – | 3.5 | 11.14 | 9.25 | 60.8 | 30.0 | 46.6 | 57.7 | 69.7 | 57.2 | 31.0 | 49.3 | 50.3 |
| | EntQuant | 1.3 | – | 3.0 | 43.77 | 54.51 | 36.2 | 23.0 | 32.5 | 49.8 | 59.1 | 57.5 | 24.0 | 6.3 | 36.1 |
| LLaMA-2 70B | Base | 16 | – | 128.5 | 5.52 | 3.32 | 82.6 | 54.4 | 65.3 | 80.2 | 81.6 | 85.4 | 49.4 | 79.3 | 72.3 |
| | Float8 | 6.5 | – | 54.9 | 5.53 | 3.33 | 82.6 | 53.9 | 65.4 | 80.2 | 81.6 | 85.0 | 49.2 | 79.2 | 72.1 |
| | EntQuant | 5.3 | – | 44.6 | 5.53 | 3.33 | 82.6 | 54.0 | 65.3 | 80.3 | 81.6 | 85.1 | 49.2 | 79.4 | 72.2 |
| | EntQuant | 4.9 | – | 40.9 | 5.55 | 3.35 | 83.0 | 54.1 | 65.1 | 80.3 | 81.3 | 85.2 | 48.8 | 79.5 | 72.2 |
| | NF4 | 4 | 64 | 36.8 | **5.59** | 3.42 | **82.5** | 53.6 | **64.9** | 79.1 | 81.3 | **85.0** | 48.2 | **79.0** | **71.7** |
| | HQQ | 4 | 64 | 36.8 | 5.60 | 3.43 | 82.2 | 53.0 | 64.5 | **79.7** | 81.2 | 84.3 | **49.0** | 78.8 | 71.6 |
| | HQQ | 4 | 128 | 34.8 | 5.63 | 3.46 | 81.9 | 53.3 | 64.5 | 79.3 | **81.4** | 84.7 | 48.8 | 78.7 | 71.6 |
| | EntQuant | 3.9 | – | **33.2** | **5.59** | **3.40** | 82.1 | **53.8** | 64.8 | 78.7 | 81.0 | **85.0** | 48.2 | 78.6 | 71.5 |
| | EntQuant | 3.5 | – | 29.9 | 5.65 | 3.48 | 82.0 | 52.8 | 64.4 | 78.5 | 81.0 | 83.7 | 48.2 | 78.5 | 71.1 |
| | HQQ | 3 | 64 | 30.5 | 6.02 | 3.95 | 80.9 | 52.4 | 63.6 | 77.3 | **80.7** | 83.9 | **47.6** | 76.7 | 70.4 |
| | HQQ | 3 | 128 | 28.5 | 6.48 | 4.35 | 79.8 | 50.4 | 61.6 | 77.0 | **80.7** | **85.0** | 45.2 | 76.2 | 69.5 |
| | EntQuant | 3 | – | **25.9** | **5.74** | **3.62** | 81.4 | **54.0** | **64.1** | **78.1** | 80.6 | 84.3 | 47.4 | **78.4** | **71.1** |
| | EntQuant | 2.5 | – | 22.1 | 6.00 | 3.94 | 80.9 | 50.9 | 62.6 | 77.1 | 80.4 | 82.8 | 46.6 | 78.1 | 69.9 |
| | HQQ | 2 | 16 | 32.9 | 32.24 | 26.16 | 67.4 | 34.1 | 41.2 | 60.1 | 70.0 | 62.3 | 34.6 | 40.6 | 51.3 |
| | HQQ | 2 | 32 | 24.9 | 323.93 | 435.23 | 29.8 | 18.4 | 27.6 | 48.9 | 55.7 | 39.9 | 23.4 | 11.1 | 31.9 |
| | HQQ | 2 | 64 | 20.9 | 2.8e3 | 3.5e3 | 26.9 | 20.6 | 26.3 | 50.7 | 53.4 | 37.8 | 26.4 | 0.8 | 30.4 |
| | EntQuant | 2.1 | – | **18.8** | **6.47** | **4.52** | **78.7** | **48.8** | **60.9** | **74.8** | **79.9** | **81.5** | **43.4** | **75.4** | **67.9** |
| | EntQuant | 1.7 | – | 15.3 | 8.43 | 6.46 | 71.6 | 40.7 | 54.8 | 68.8 | 76.0 | 69.4 | 39.8 | 60.6 | 60.2 |
| | EntQuant | 1.3 | – | 12.7 | 690.33 | 413.31 | 48.9 | 25.2 | 34.9 | 53.0 | 60.3 | 37.9 | 28.6 | 7.2 | 37.0 |

*Table C.3.* Full baseline results for the LLaMA-3.1 base model family (8B, 70B). We report perplexity on C4 and WikiText-2, as well as zero-shot accuracy on eight tasks from LM Eval. All results are generated in-house using the configurations described in Section A. Best results per model and bit-rate group are in **bold**. This table extends the results presented in Table 3.

| Model | Method | Bits | Group | Size (GiB) | C4 ↓ | WikiText-2 ↓ | ARC-E ↑ | ARC-C ↑ | HellaSwag ↑ | WinoGrande ↑ | PIQA ↑ | BoolQ ↑ | OBQA ↑ | LAMBADA ↑ | LM Eval Avg. ↑ |
|---|---|---|---|---|---|---|---|---|---|---|---|---|---|---|---|
| LLaMA-3.1 8B | Base | 16 | – | 15.0 | 8.43 | 5.84 | 82.0 | 52.0 | 60.8 | 74.3 | 79.4 | 83.0 | 45.4 | 74.8 | 68.9 |
| | NF4 | 4 | 64 | 5.6 | 8.96 | 6.22 | 81.1 | **51.5** | 59.5 | 73.5 | 78.7 | **82.3** | 44.6 | 73.4 | 68.1 |
| | HQQ | 4 | 64 | 5.6 | 9.02 | 6.26 | 81.2 | 50.9 | 59.7 | **73.7** | **79.6** | 81.9 | **45.4** | **74.5** | **68.4** |
| | HQQ | 4 | 128 | **5.4** | 9.19 | 6.37 | 80.2 | 48.4 | 59.0 | 73.6 | 78.3 | 81.8 | 44.4 | 73.3 | 67.4 |
| | EntQuant | 4 | – | 5.5 | **8.76** | **6.08** | 81.9 | 51.5 | **60.1** | 73.3 | 79.1 | **82.3** | 44.4 | 73.7 | 68.3 |
| | EntQuant | 3.5 | – | 5.1 | 9.06 | 6.28 | 80.1 | 49.6 | 59.4 | 72.2 | 78.3 | 82.4 | 44.8 | 73.3 | 67.5 |
| | HQQ | 3 | 64 | 5.0 | 12.14 | 8.60 | 74.6 | 42.7 | 53.2 | 70.9 | 76.3 | 72.9 | 38.6 | 64.2 | 61.7 |
| | HQQ | 3 | 128 | 4.8 | 14.46 | 10.31 | 70.1 | 35.8 | 51.1 | 67.5 | 74.2 | 67.2 | 38.6 | 60.8 | 58.2 |
| | EntQuant | 3 | – | **4.7** | **9.74** | **6.77** | **78.2** | **47.5** | **57.9** | **71.7** | **78.0** | **80.5** | **42.6** | **71.8** | **66.0** |
| | EntQuant | 2.6 | – | 4.4 | 11.25 | 7.87 | 77.6 | 44.3 | 55.5 | 69.4 | 77.8 | 70.7 | 42.6 | 67.5 | 63.2 |
| | HQQ | 2 | 16 | 5.2 | 142.94 | 163.36 | 33.0 | 19.8 | 28.4 | 50.0 | 57.7 | 52.9 | 24.0 | 2.6 | 33.6 |
| | HQQ | 2 | 32 | 4.4 | 1.9e3 | 2.9e3 | 25.7 | 19.6 | 26.6 | 52.3 | 53.2 | 37.8 | 24.0 | 0.1 | 29.9 |
| | HQQ | 2 | 64 | **4.0** | 1.8e4 | 3.6e4 | 27.0 | 22.1 | 25.7 | 49.6 | 53.6 | 40.1 | 27.2 | 0.0 | 30.7 |
| | EntQuant | 2.1 | – | **4.0** | **17.56** | **13.86** | **60.0** | **31.2** | **47.4** | **62.4** | **71.1** | **60.9** | **37.6** | **50.1** | **52.6** |
| | EntQuant | 1.7 | – | 3.7 | 135.24 | 186.03 | 40.0 | 21.2 | 34.0 | 53.7 | 61.9 | 42.1 | 27.2 | 12.1 | 36.8 |
| | EntQuant | 1.4 | – | 3.4 | 6.4e3 | 3.0e4 | 26.3 | 22.7 | 26.2 | 51.3 | 54.0 | 38.1 | 25.2 | 0.0 | 30.5 |
| LLaMA-3.1 70B | Base | 16 | – | 131.4 | 5.82 | 2.64 | 85.0 | 59.3 | 67.5 | 81.8 | 82.5 | 87.2 | 47.4 | 79.4 | 73.8 |
| | NF4 | 4 | 64 | 39.8 | 6.35 | 3.07 | 85.8 | 57.9 | 66.7 | 80.7 | **82.9** | 85.9 | 45.8 | **78.8** | 73.1 |
| | HQQ | 4 | 64 | 39.8 | 6.61 | 3.07 | 85.6 | 58.4 | 65.9 | 80.3 | 82.5 | 84.6 | 46.6 | 77.7 | 72.7 |
| | HQQ | 4 | 128 | 37.8 | 7.08 | 3.47 | 83.3 | 56.1 | 66.0 | 80.6 | 82.1 | 85.3 | **47.6** | 78.1 | 72.4 |
| | EntQuant | 3.9 | – | **36.2** | **6.06** | **2.95** | **85.9** | **59.3** | **67.1** | **81.1** | 82.6 | **87.0** | 47.4 | 77.9 | **73.6** |
| | EntQuant | 3.5 | – | 32.9 | 6.26 | 3.21 | 84.8 | 58.3 | 66.6 | 80.7 | 82.2 | 86.4 | 49.0 | 77.9 | 73.2 |
| | HQQ | 3 | 64 | 33.4 | 629.09 | 327.44 | 54.9 | 22.1 | 29.1 | 54.5 | 64.7 | 47.6 | 31.8 | 7.2 | 39.0 |
| | HQQ | 3 | 128 | 31.4 | 620.89 | 996.49 | 42.8 | 20.6 | 27.5 | 55.8 | 58.1 | 38.2 | 28.0 | 2.0 | 34.1 |
| | EntQuant | 3 | – | **28.9** | **6.76** | **3.76** | **84.2** | **58.5** | **66.0** | **80.0** | **81.8** | **86.4** | 46.6 | **77.1** | **72.6** |
| | EntQuant | 2.5 | – | 25.0 | 7.67 | 4.77 | 82.2 | 55.2 | 63.7 | 78.3 | 81.9 | 84.1 | 46.4 | 77.7 | 71.2 |
| | HQQ | 2 | 16 | 35.8 | 7.7e3 | 8.4e3 | 27.9 | 18.0 | 26.5 | 49.8 | 53.9 | 37.8 | 25.8 | 0.3 | 30.0 |
| | HQQ | 2 | 32 | 27.8 | 3.0e4 | 3.3e4 | 25.0 | 18.9 | 25.7 | 51.5 | 52.7 | 37.9 | 27.6 | 0.1 | 29.9 |
| | HQQ | 2 | 64 | 23.8 | 1.3e4 | 2.0e4 | 25.7 | 21.2 | 25.8 | 50.6 | 52.4 | 37.8 | 25.4 | 0.0 | 29.9 |
| | EntQuant | 2.1 | – | **21.7** | **9.92** | **6.16** | **80.6** | **52.3** | **61.3** | **75.7** | **80.6** | **80.6** | **45.8** | **71.6** | **68.6** |
| | EntQuant | 1.7 | – | 18.2 | 7.9e3 | 1.6e4 | 40.1 | 20.1 | 28.7 | 52.1 | 59.5 | 50.4 | 27.8 | 5.4 | 35.5 |
| | EntQuant | 1.3 | – | 15.6 | 2.9e4 | 4.8e4 | 25.4 | 21.9 | 25.8 | 49.5 | 51.6 | 38.3 | 30.0 | 0.0 | 30.3 |

# D. Full Results for Calibration and Fine-Tuning Methods

*Table D.1.* Comparison of EntQuant with calibration and fine-tuning methods on the LLaMA-2 base model family (7B, 13B, 70B). We report perplexity on C4 and WikiText-2, and zero-shot accuracy on five tasks from LM Eval. Results for GPTQ (Frantar et al., 2023), AWQ (Lin et al., 2024), OmniQuant (Shao et al., 2024), LeanQuant (Zhang & Shrivastava, 2025), SqueezeLLM (Kim et al., 2024), AQLM (Egiazarian et al., 2024), QuIP# (Tseng et al., 2024), and EfficientQAT (Chen et al., 2025) are taken from Chen et al. (2025) and Zhang & Shrivastava (2025), with additions from Tseng et al. (2024) and Shao et al. (2024) for missing perplexity scores. Note that, at 2.1 bits, EntQuant operates below the overhead introduced by a group size of 128, which yields 2.14 bits per parameter (Chen et al., 2025, Table 11). Best results per model and bit-rate group are in **bold**. This table extends Table 4 to include additional model sizes and comparison methods.

| Model | Method | Bits | Group | C4 ↓ | WikiText-2 ↓ | ARC-E ↑ | ARC-C ↑ | HellaSwag ↑ | WinoGrande ↑ | PIQA ↑ | LM Eval Avg. ↑ |
|---|---|---|---|---|---|---|---|---|---|---|---|
| LLaMA-2 7B | Base | 16 | – | 6.98 | 5.47 | 75.5 | 42.5 | 57.1 | 69.7 | 77.7 | 64.5 |
| | EntQuant | 3 | – | 7.55 | 5.95 | 74.6 | **42.9** | **56.3** | 68.4 | **77.7** | **64.0 (-0.8%)** |
| | GPTQ | 3 | 128 | 7.89 | 6.29 | 73.7 | 40.2 | 53.7 | 68.6 | 76.0 | 62.4 (-3.2%) |
| | AWQ | 3 | 128 | 7.84 | 6.24 | 74.1 | 41.6 | 55.0 | 67.4 | 76.0 | 62.8 (-2.6%) |
| | OmniQuant | 3 | 128 | 7.75 | 6.03 | 74.4 | 39.9 | 54.4 | 66.7 | 76.8 | 62.4 (-3.2%) |
| | LeanQuant$_{nu}$ | 3 | – | 7.73 | 6.19 | 73.7 | 40.2 | 53.2 | 68.3 | 76.4 | 62.4 (-3.3%) |
| | SqueezeLLM | 3 | – | 7.72 | 6.18 | 73.1 | 40.3 | 54.1 | 67.9 | 76.5 | 62.4 (-3.3%) |
| | QuIP# | 3 | – | **7.32** | **5.79** | 74.6 | 41.9 | 55.9 | 68.2 | 77.0 | 63.5 (-1.5%) |
| | EfficientQAT | 3 | 128 | 7.34 | 5.81 | **74.7** | 42.8 | 55.9 | **69.1** | 77.6 | **64.0 (-0.8%)** |
| | EntQuant | 2.1 | – | 10.59 | 8.29 | 68.9 | 36.2 | 51.6 | 65.7 | 73.7 | 59.2 (-8.3%) |
| | GPTQ | 2 | 128 | 33.70 | 36.77 | 40.5 | 21.2 | 32.6 | 55.2 | 58.3 | 41.6 (-35.6%) |
| | OmniQuant | 2 | 128 | 15.02 | 11.06 | 50.1 | 23.5 | 40.3 | 55.9 | 65.1 | 47.0 (-27.2%) |
| | LeanQuant$_{nu}$ | 2 | – | 17.07 | 15.51 | 51.8 | 24.0 | 35.9 | 58.2 | 66.4 | 47.2 (-26.8%) |
| | AQLM | 2 | 1x16 | – | – | **74.1** | **39.7** | **53.4** | 65.2 | **76.9** | **61.8 (-4.1%)** |
| | QuIP# | 2 | – | **8.35** | **6.66** | 71.8 | 37.9 | 52.2 | 65.7 | 75.5 | 60.6 (-6.1%) |
| | EfficientQAT | 2 | 64 | 8.50 | 6.86 | 71.0 | 36.9 | 51.6 | 66.0 | 75.3 | 60.1 (-6.8%) |
| | EfficientQAT | 2 | 128 | 8.79 | 7.19 | 69.8 | 36.5 | 50.8 | **66.2** | 74.2 | 59.5 (-7.8%) |
| LLaMA-2 13B | Base | 16 | – | 6.47 | 4.88 | 78.9 | 47.2 | 60.2 | 72.2 | 79.7 | 67.6 |
| | EntQuant | 3 | – | 6.76 | 5.15 | 77.3 | 44.3 | **59.4** | 72.1 | 78.5 | 66.3 (-2.0%) |
| | GPTQ | 3 | 128 | 7.00 | 5.42 | 78.0 | 45.6 | 57.8 | 70.9 | **78.6** | 66.2 (-2.1%) |
| | AWQ | 3 | 128 | 6.94 | 5.32 | 78.0 | 44.6 | 58.6 | 71.8 | 77.8 | 66.1 (-2.2%) |
| | OmniQuant | 3 | 128 | 6.98 | 5.28 | 77.9 | 46.2 | 58.5 | 70.0 | 78.4 | 66.2 (-2.2%) |
| | LeanQuant$_{nu}$ | 3 | – | 6.98 | 5.40 | 77.2 | 44.2 | 56.4 | 70.1 | 77.8 | 65.1 (-3.7%) |
| | SqueezeLLM | 3 | – | 6.97 | 5.36 | 77.3 | 43.2 | 58.7 | 69.5 | 77.9 | 65.3 (-3.4%) |
| | QuIP# | 3 | – | **6.72** | **5.10** | 77.9 | 44.6 | 58.3 | **72.5** | 78.1 | 66.3 (-2.0%) |
| | EfficientQAT | 3 | 128 | 6.73 | 5.12 | **79.0** | **48.0** | 59.0 | 72.1 | 78.4 | **67.3 (-0.5%)** |
| | EntQuant | 2.1 | – | 7.95 | 6.24 | 74.5 | 41.4 | 55.6 | 69.6 | 76.9 | 63.6 (-6.0%) |
| | GPTQ | 2 | 128 | 20.97 | 28.14 | 55.6 | 21.9 | 41.1 | 55.8 | 67.1 | 48.3 (-28.6%) |
| | OmniQuant | 2 | 128 | 11.05 | 8.26 | 63.2 | 30.3 | 46.2 | 57.9 | 70.1 | 53.6 (-20.8%) |
| | LeanQuant$_{nu}$ | 2 | – | 11.83 | 10.06 | 62.5 | 30.2 | 42.2 | 62.0 | 69.9 | 53.4 (-21.1%) |
| | AQLM | 2 | 1x16 | – | – | 75.2 | **43.5** | 57.6 | 70.1 | 78.3 | **65.0 (-4.0%)** |
| | QuIP# | 2 | – | **7.45** | **5.74** | 75.7 | 42.9 | 56.5 | 69.1 | 78.0 | 64.4 (-4.7%) |
| | EfficientQAT | 2 | 64 | 7.59 | 5.96 | 74.8 | 41.9 | 55.3 | 68.4 | 77.0 | 63.5 (-6.1%) |
| | EfficientQAT | 2 | 128 | 7.75 | 6.08 | 75.0 | 42.8 | 55.7 | 68.9 | 77.0 | 63.9 (-5.5%) |
| LLaMA-2 70B | Base | 16 | – | 5.52 | 3.32 | 82.6 | 54.4 | 65.3 | 80.2 | 81.6 | 72.8 |
| | EntQuant | 3 | – | 5.74 | 3.62 | 81.4 | 54.0 | 64.1 | **78.1** | 80.6 | 71.7 (-1.6%) |
| | GPTQ | 3 | 128 | 5.85 | 3.85 | 81.7 | 53.7 | 62.9 | 77.7 | 81.5 | 71.5 (-1.9%) |
| | AWQ | 3 | 128 | 5.81 | 3.74 | 81.4 | 53.7 | 63.7 | 76.5 | 81.8 | 71.4 (-1.9%) |
| | OmniQuant | 3 | 128 | 5.85 | 3.78 | 81.0 | 52.8 | 63.5 | 76.5 | 81.5 | 71.1 (-2.4%) |
| | QuIP# | 3 | – | **5.67** | **3.56** | 82.1 | 55.9 | 64.2 | 76.2 | 82.2 | 72.1 (-0.9%) |
| | EfficientQAT | 3 | 128 | 5.71 | 3.61 | 81.7 | 53.8 | 64.2 | 77.3 | 81.8 | 71.8 (-1.5%) |
| | EntQuant | 2.1 | – | 6.47 | 4.52 | 78.7 | 48.8 | 60.9 | 74.8 | 79.9 | 68.6 (-5.8%) |
| | GPTQ | 2 | 128 | – | – | 25.1 | 22.7 | 25.0 | 49.6 | 49.5 | 34.4 (-52.8%) |
| | OmniQuant | 2 | 128 | 8.52 | 6.55 | 67.2 | 33.3 | 35.5 | 64.3 | 74.1 | 54.9 (-24.6%) |
| | AQLM | 2 | 1x16 | – | – | 81.4 | **53.0** | 62.8 | **76.0** | 81.1 | 70.8 (-2.7%) |
| | QuIP# | 2 | – | **6.12** | **4.16** | 81.9 | 52.6 | **62.9** | 75.8 | **81.4** | 70.9 (-2.6%) |
| | EfficientQAT | 2 | 64 | 6.38 | 4.52 | 80.1 | 50.8 | 61.8 | 74.6 | 80.1 | 69.5 (-4.6%) |
| | EfficientQAT | 2 | 128 | 6.48 | 4.61 | 80.0 | 49.2 | 61.6 | 73.6 | 80.2 | 68.9 (-5.3%) |

# E. Full Results for Instruction-Tuned Models

*Table E.1.* Full results on a variety of instruction-tuned models across four advanced benchmarks from LM Eval: instruction-following (IFEval), mathematical reasoning (GSM8K with chain-of-thought), scientific reasoning (GPQA), and broad knowledge (MMLU); see Section A.4 for more details on the benchmark configurations. These results extend the visualization of Figure 1.

| Model | Method | Bits | C4 ↓ | WikiText-2 ↓ | GSM8K ↑ | GPQA ↑ | MMLU ↑ | IFEval ↑ | LM Eval Avg. ↑ |
|---|---|---|---|---|---|---|---|---|---|
| LLaMA-3.1 8B Instruct | Base | 16 | 9.75 | 6.76 | 84.7 | 30.8 | 68.7 | 73.6 | 64.4 |
| | EntQuant | 4 | 10.11 | 7.02 | 83.7 | 29.9 | 67.5 | 76.0 | 64.3 |
| | EntQuant | 3 | 11.11 | 7.73 | 75.1 | 28.6 | 63.9 | 70.8 | 59.6 |
| | EntQuant | 2.2 | 17.95 | 13.57 | 14.6 | 24.6 | 48.6 | 49.0 | 34.2 |
| LLaMA-3.1 70B Instruct | Base | 16 | 6.74 | 3.59 | 94.6 | 41.5 | 83.1 | 84.5 | 75.9 |
| | EntQuant | 3.9 | 7.04 | 3.83 | 94.5 | 42.0 | 82.6 | 85.0 | 76.0 |
| | EntQuant | 3 | 7.72 | 4.51 | 93.3 | 37.9 | 81.3 | 82.6 | 73.8 |
| | EntQuant | 2.1 | 11.23 | 6.76 | 87.4 | 37.7 | 75.6 | 78.0 | 69.7 |
| LLaMA-3.3 70B Instruct | Base | 16 | 6.87 | 3.62 | 94.3 | 50.4 | 82.0 | 88.9 | 78.9 |
| | EntQuant | 3.9 | 7.19 | 3.94 | 93.9 | 50.9 | 81.9 | 87.6 | 78.6 |
| | EntQuant | 3 | 7.89 | 4.63 | 93.7 | 44.2 | 80.6 | 87.6 | 76.5 |
| | EntQuant | 2.1 | 11.29 | 6.97 | 86.7 | 35.7 | 75.7 | 86.1 | 71.1 |
| Qwen3 8B | Base | 16 | 13.30 | 9.73 | 81.4 | 39.3 | 72.0 | 81.7 | 68.6 |
| | EntQuant | 3.9 | 13.62 | 9.96 | 76.0 | 37.5 | 69.6 | 82.3 | 66.3 |
| | EntQuant | 3 | 14.76 | 11.24 | 75.4 | 34.4 | 63.3 | 78.7 | 63.0 |
| | EntQuant | 2.2 | 20.12 | 17.64 | 28.7 | 23.2 | 30.3 | 56.9 | 34.8 |
| Qwen3 14B | Base | 16 | 12.02 | 8.64 | 82.8 | 39.3 | 76.7 | 84.5 | 70.8 |
| | EntQuant | 3.9 | 12.29 | 8.87 | 80.2 | 37.5 | 76.5 | 84.3 | 69.6 |
| | EntQuant | 3 | 12.70 | 9.29 | 82.8 | 31.5 | 74.3 | 84.3 | 68.2 |
| | EntQuant | 2.2 | 14.98 | 11.61 | 68.0 | 28.8 | 57.9 | 76.3 | 57.8 |
| Qwen3 32B | Base | 16 | 10.78 | 7.61 | 89.2 | 42.6 | 80.9 | 84.3 | 74.2 |
| | EntQuant | 3.9 | 10.98 | 7.79 | 86.3 | 43.3 | 80.4 | 82.8 | 73.2 |
| | EntQuant | 3 | 11.32 | 8.13 | 84.7 | 39.7 | 78.2 | 80.8 | 70.9 |
| | EntQuant | 2.2 | 13.47 | 10.45 | 79.9 | 37.3 | 71.7 | 78.7 | 66.9 |
| OLMo-3.1 32B Instruct | Base | 16 | 11.52 | 6.95 | 93.9 | 37.7 | 74.1 | 87.4 | 73.3 |
| | EntQuant | 3.9 | 11.59 | 7.05 | 93.7 | 37.5 | 73.8 | 87.2 | 73.1 |
| | EntQuant | 3 | 11.66 | 7.21 | 93.3 | 38.8 | 73.3 | 88.0 | 73.4 |
| | EntQuant | 2.1 | 12.71 | 8.43 | 93.4 | 35.5 | 70.4 | 84.3 | 70.9 |
| Mistral Large Instruct 24.11 123B | Base | 16 | 5.53 | 2.64 | 84.2 | 47.5 | 83.1 | 81.9 | 74.2 |
| | EntQuant | 3.9 | 5.60 | 2.73 | 85.4 | 45.8 | 82.9 | 81.7 | 73.9 |
| | EntQuant | 3 | 5.80 | 2.99 | 84.7 | 46.2 | 82.4 | 79.3 | 73.1 |
| | EntQuant | 2.1 | 6.63 | 4.01 | 82.6 | 42.9 | 76.8 | 78.7 | 70.3 |

# F. Full Results for Model Inference

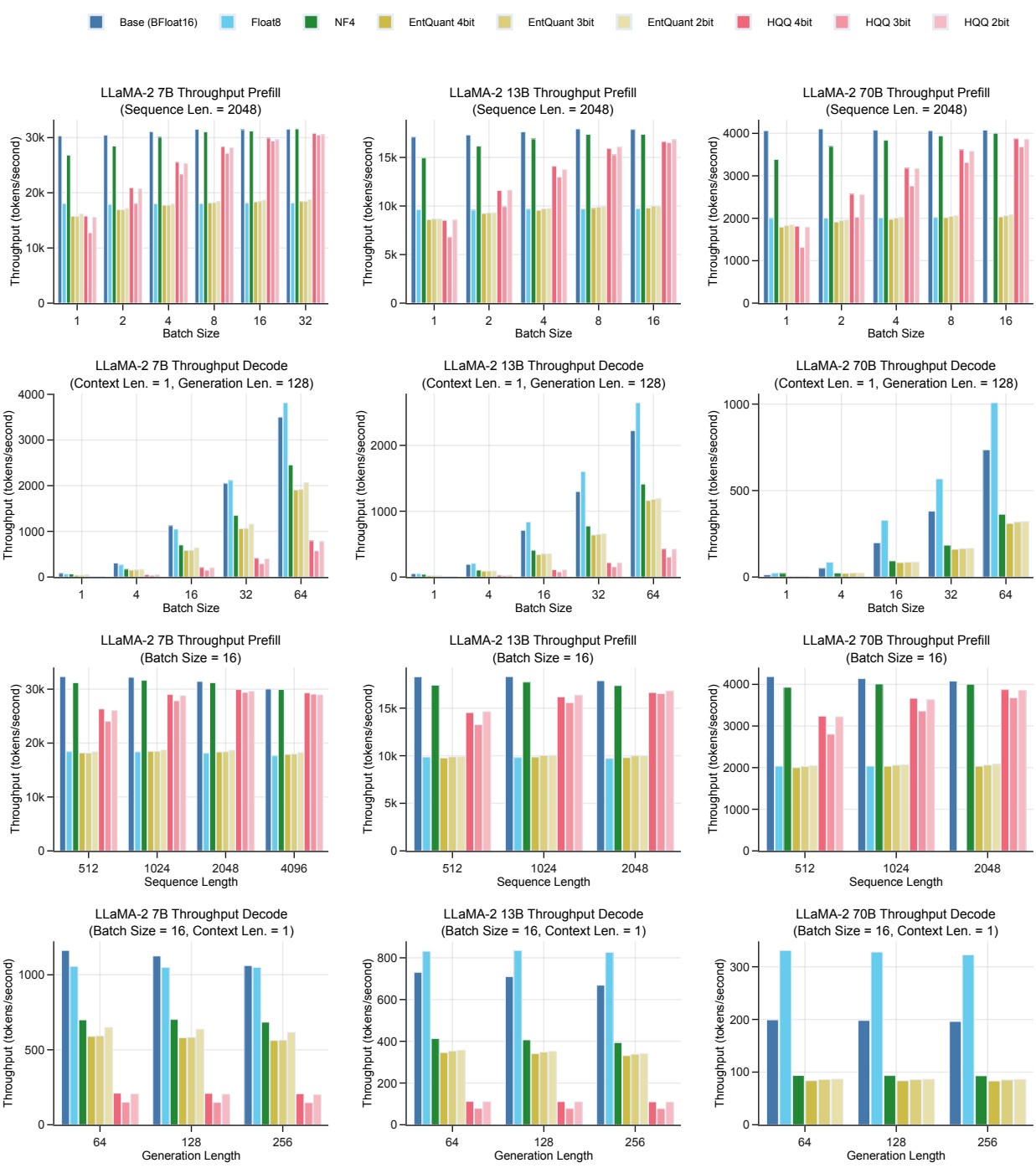

*Figure F.1.* Inference **throughput** comparison across the LLaMA-2 model base family (7B, 13B, 70B) under varying inference configurations. Top two rows: Prefill and decode throughput as a function of batch size. Bottom two rows: Prefill and decode throughput as a function of sequence/generation length. Missing values are due to CUDA out-of-memory errors and/or exceptionally long runtimes. EntQuant benefits from the Marlin kernel's superior performance, particularly in pure decoding regimes (input context length 1). In prefill-dominated regimes, both Float8 and EntQuant are slightly slower than the BFloat16 baseline due to kernel overhead. These results extend Figure 5.

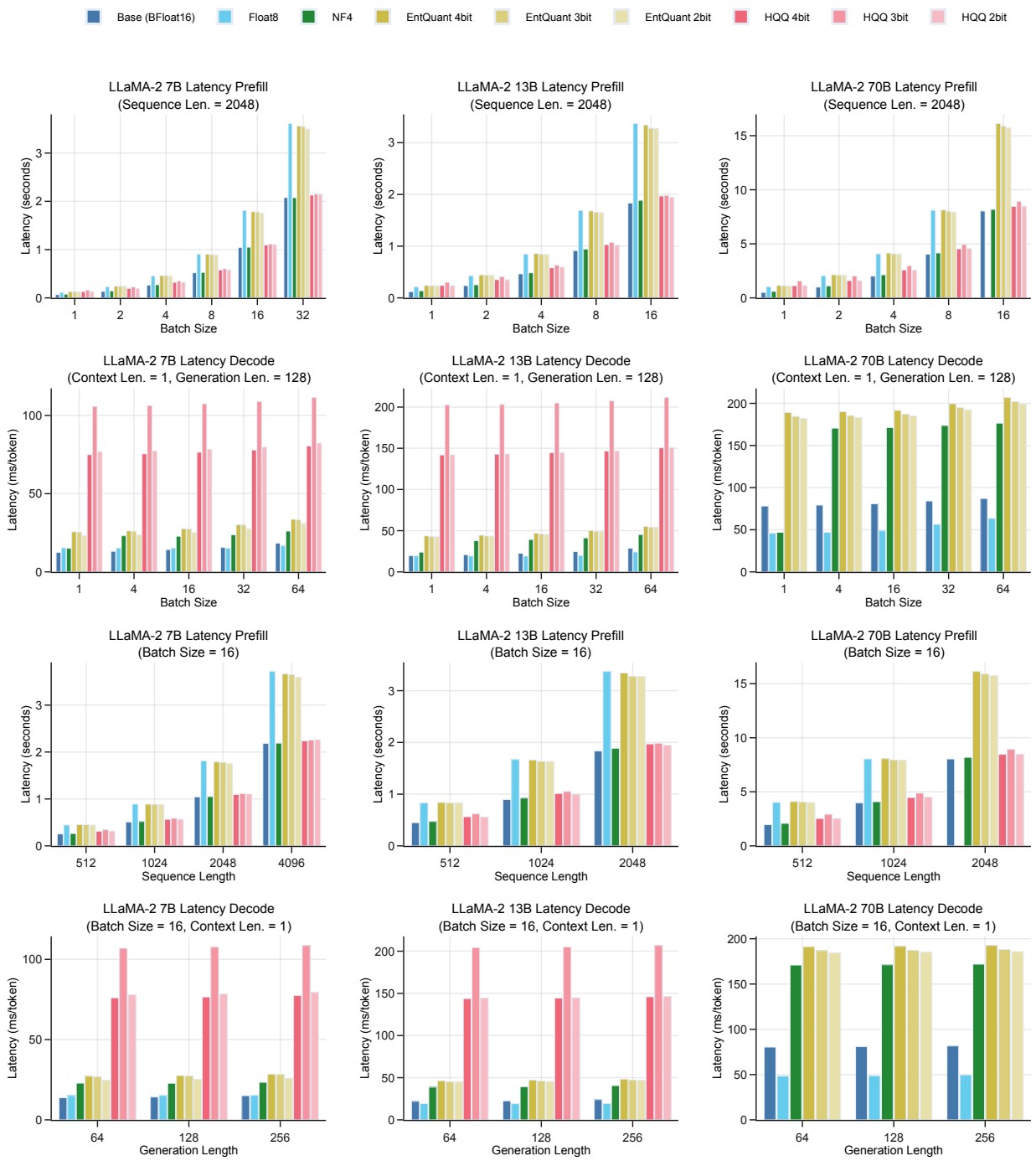

*Figure F.2.* Inference **latency** comparison across the LLaMA-2 model base family (7B, 13B, 70B) under varying inference configurations. Top two rows: Prefill and decode latency as a function of batch size. Bottom two rows: Prefill and decode latency as a function of sequence/generation length. Missing values are due to CUDA out-of-memory errors and/or exceptionally long runtimes. These results extend Figure 5.

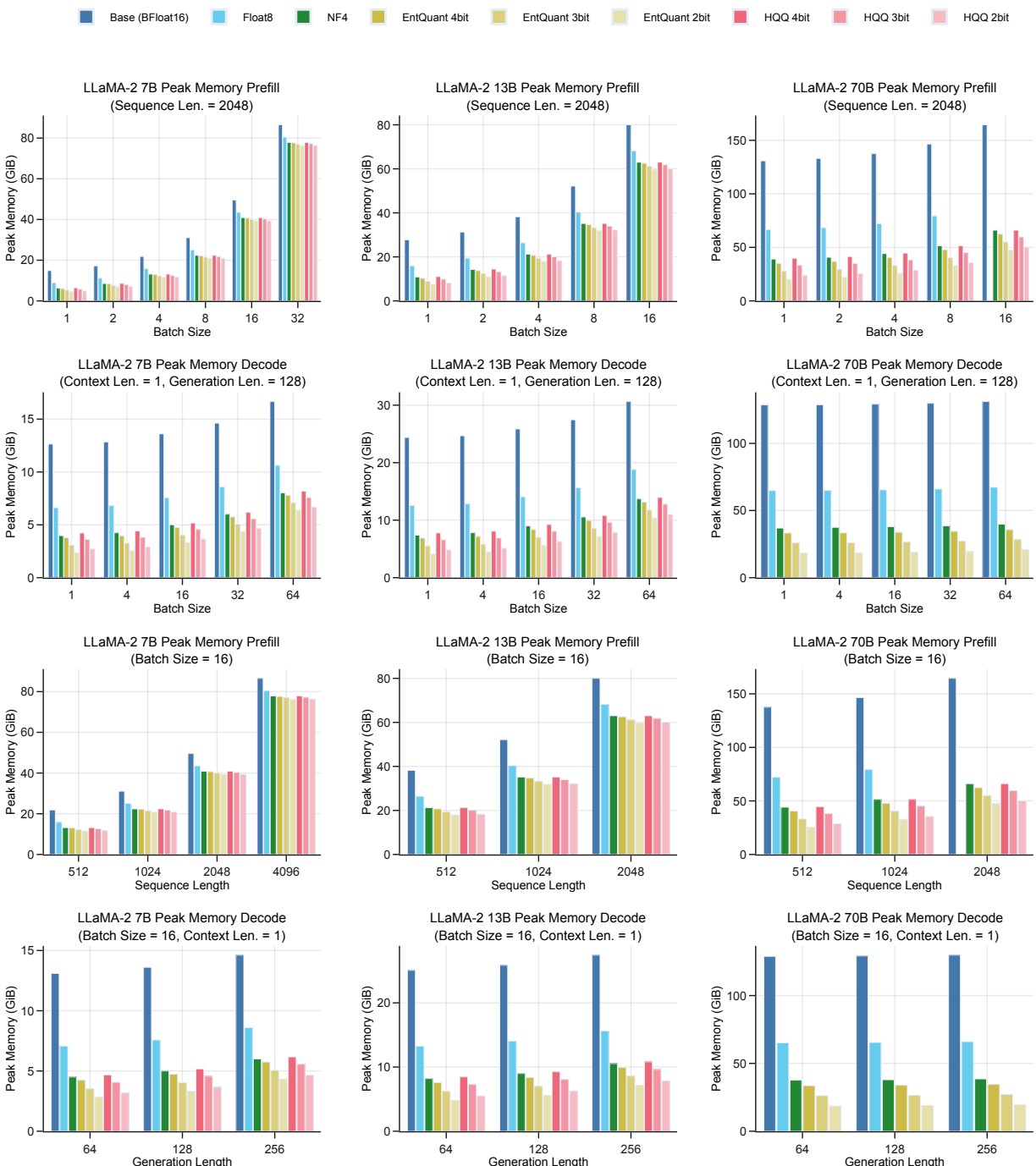

*Figure F.3.* Inference **peak memory** comparison across the LLaMA-2 model base family (7B, 13B, 70B) under varying inference configurations. Top two rows: Prefill and decode peak memory as a function of batch size. Bottom two rows: Prefill and decode peak memory as a function of sequence/generation length. Missing values are due to CUDA out-of-memory errors and/or exceptionally long runtimes. These results extend Figure 5.

# G. Full Results for EntQuant Float8 vs. Int8 and Super Weight Handling

*Table G.1.* Comparison of EntQuant with Float8 and Int8, including an ablation study on super weight handling for the LLaMA-2 base model family (7B, 13B, 70B). We compare EntQuant with Float8 and Int8 base formats under different super weight (SW) exclusion thresholds: 10, 50, and $\infty$ (no exclusion). Lower thresholds declare more weights as super weights, leading to more excluded layers; see Section 3.5 for more details. For comparison, we also show NF4 and HQQ results with and without super weight exclusion (using threshold 50, which is the default in Yu et al. (2024)). These results extend the analysis of Figure 7.

| Method | Bits | Group | C4 ↓ (Perplexity) | | | WikiText-2 ↓ (Perplexity) | | |
|---|---|---|---|---|---|---|---|---|
| | | | 2-7 | 2-13 | 2-70 | 2-7 | 2-13 | 2-70 |
| Base | 16 | – | 6.98 | 6.47 | 5.52 | 5.47 | 4.88 | 3.32 |
| NF4 | 4 | 64 | 7.16 | 6.57 | 5.59 | 5.65 | 4.98 | 3.42 |
| NF4 (SW) | 4 | 64 | 7.10 | 6.55 | 5.57 | 5.59 | 4.97 | 3.40 |
| HQQ | 4 | 64 | 7.20 | 6.57 | 5.60 | 5.68 | 4.98 | 3.43 |
| HQQ (SW) | 4 | 64 | 7.10 | 6.55 | 5.57 | 5.60 | 4.96 | 3.39 |
| EntQuant Float8 (SW 10) | 4 | – | 7.07 | 6.53 | 5.59 | 5.56 | 4.95 | 3.39 |
| EntQuant Float8 (SW 50) | 3.9/4 | – | 7.07 | 6.53 | 5.59 | 5.57 | 4.96 | 3.39 |
| EntQuant Float8 (SW Inf) | 3.9 | – | 7.14 | 6.55 | 5.59 | 5.62 | 4.97 | 3.40 |
| EntQuant Int8 (SW 10) | 4 | – | 7.07 | 6.53 | 5.57 | 5.56 | 4.96 | 3.38 |
| EntQuant Int8 (SW 50) | 3.9/4 | – | 7.08 | 11.54 | 5.57 | 5.56 | 7.82 | 3.39 |
| EntQuant Int8 (SW Inf) | 3.9 | – | 21.42 | 13.13 | 5.97 | 22.10 | 9.55 | 3.83 |
| HQQ | 3 | 64 | 9.11 | 7.29 | 6.02 | 7.05 | 5.60 | 3.95 |
| HQQ (SW) | 3 | 64 | 7.76 | 6.93 | 5.86 | 6.16 | 5.33 | 3.77 |
| EntQuant Float8 (SW 10) | 3.1 | – | 7.41 | 6.72 | 5.71 | 5.86 | 5.12 | 3.57 |
| EntQuant Float8 (SW 50) | 3/3.1 | – | 7.45 | 6.73 | 5.71 | 5.87 | 5.12 | 3.57 |
| EntQuant Float8 (SW Inf) | 3 | – | 7.55 | 6.76 | 5.74 | 5.95 | 5.15 | 3.62 |
| EntQuant Int8 (SW 10) | 3.1 | – | 7.41 | 6.72 | 5.68 | 5.86 | 5.12 | 3.55 |
| EntQuant Int8 (SW 50) | 3/3.1 | – | 7.45 | 6.73 | 5.68 | 5.87 | 5.14 | 3.56 |
| EntQuant Int8 (SW Inf) | 3 | – | 7.61 | 6.75 | 5.76 | 5.99 | 5.16 | 3.63 |
| HQQ | 2 | 64 | 6.0e3 | 2.7e3 | 2.8e3 | 7.5e3 | 3.8e3 | 3.5e3 |
| HQQ (SW) | 2 | 64 | 910.97 | 357.99 | 2.4e3 | 1.2e3 | 395.38 | 3.0e3 |
| EntQuant Float8 (SW 10) | 2.2 | – | 9.75 | 7.98 | 6.34 | 7.84 | 6.26 | 4.44 |
| EntQuant Float8 (SW 50) | 2.1 | – | 10.67 | 8.05 | 6.35 | 8.44 | 6.36 | 4.46 |
| EntQuant Float8 (SW Inf) | 2 | – | 11.33 | 8.10 | 6.59 | 8.84 | 6.39 | 4.66 |
| EntQuant Int8 (SW 10) | 2.2 | – | 9.74 | 7.96 | 6.33 | 7.82 | 6.24 | 4.43 |
| EntQuant Int8 (SW 50) | 2.1 | – | 10.65 | 8.03 | 6.34 | 8.42 | 6.33 | 4.44 |
| EntQuant Int8 (SW Inf) | 2 | – | 11.31 | 8.09 | 6.55 | 8.81 | 6.36 | 4.62 |

