# OpenReview forum: "Float8@2bits: Entropy Coding Enables Data-Free Model Compression"
_ICML.cc/2026/Conference — ICML 2026 regular_

### Official Review · Reviewer_ZKAA · 2026-03-08

**Soundness:** 3
**Presentation:** 1
**Significance:** 3
**Originality:** 3
**Overall Recommendation:** 4
**Confidence:** 4

**Summary:**

This paper introduces EntQuant, a post-training compression framework that combines quantization with lossless entropy coding to achieve effective very small bitrates in the extreme compression scope.  The main idea is that we do not have to force weights into strict 2-bit or 4-bit formats. Instead, EntQuant optimizes Float8/Int8 quantized weights for low entropy and then compresses them with ANS. The proposed method is  data-free and model-agnostic, it does not need calibration data or extra training.

The idea is interesting and making the entropy coding actually work inside the inference pipeline (rather than pure offline storage) demonstrates non-trivial engineering effort.  However, some important practical details are only in the appendix, which makes the main paper a bit hard to follow.

**Compliance With Llm Reviewing Policy:**

Affirmed.

**Final Justification:**

The paper introduces a novel pipeline that decouples the quantization bit-width and compression. The proposed method is sound and has potential benefits for deployment of large models.

The authors’ rebuttal addressed my clarification questions. They would also like to add longer context evaluation. I would like to raise my score to “accept”.

**Key Questions For Authors:**

The questions are similar to the weakness.
(1) Please state clearly in the main paper about the decompression pipeline, and what temporary buffer size is required.

(2) Can the authors provide a table in the main paper to report the full inference-time memory footprint. It is a strong support to include compressed weight storage, scales, metadata, decompression buffer, peak model memory footprint during inference and KV cache.

(3) Can the authors expand the discussion of speed to include longer context and generation length in the main-paper? The LLaMa 2 supports up to 8192 input tokens.

**Limitations:**

Yes for technical limitations. The authors adequately discuss the technical limitations of the current method, including simplified design choices, such as the use of an L1-based regularizer instead of more complex grouping structures, as well as the absence of fused decompression kernels. The paper also makes clear that computational budget constraints limit the choice of model sizes and benchmark datasets.

No, the paper lacks societal impact discussion. Since the method enables stronger models to run on lower-resource hardware, it has positive societal benefits by reducing deployment cost. On the other hand, it may also lower the barrier to deployment where safeguards are insufficient. The paper would be strengthened by discussing these tradeoffs. For example, adding potential future work to evaluate   safety-relevant behaviors (toxicity or harmful-content generation) related to compression method will be helpful.

**Strengths And Weaknesses:**

Strengths
(1)The key contribution is to decouple compression rate from fixed quantization bit-width. The proposed method optimizes quantized weights under the entropy constraints and applies ANS-based coding. This is a novel and interesting idea to overcome the limit of low-bit PTQ for extreme compression below 4 bits.

(2) The method only requires weights, without calibration data or retraining. This is practically useful for instruction-tuned models where training data may be unavailables.

(3) The evaluation runs over 480 runs across 16 open-weight LLMs including base and instruction-tuned models. The experiment reports perplexity, zero-shot LM Eval and more challenging instruction-following/reasoning benchmarks. This is a very impressive experimental effort.

(4) The results shows promising results that EntQuant substantially outperforms common data-free baselines in the in the 2–3 bit extreme compression scope, where HQQ tends to fail.

(5) In the appendix, the authors clarify that the implementation uses a block-wise decompression buffer, the weights of one transformer block is compressed into a single bitstream and decompressed before the block forward pass. This on-the-fly decompression idea is a good contribution to systems research.

Weakness
(1) Important implementation details are missing from the main paper. Algorithm 2 in the main text explains the decompression pipeline at a high level. However, important practical question about how decompression is implemented for inference is not clear. Although the appendix later explains with more details, this should be in the main paper because it is key for deployment.

(2) The paper explains that the memory footprint includes the compressed bitstream, scales, and decoder metadata, and points to the appendix figures for peak-memory usage. Again, the main paper does not provide the picture of runtime footprint. We need to see the decompression buffer size and peak memory more clearly.

(3) The paper reports that the promising throughput of EntQuant is only 1.5x~2x slower than the BFloat16 baseline and similar to NF4. However, the experiments are only on 512 input and 256 output tokens. Since the method introduces the on-the-fly decompression, it would be useful to see  longer-context and longer-generation performance in the main paper.

---

> ### Author Rebuttal · Authors · 2026-03-27
>
> We thank Reviewer ZKAA for the thorough and positive review. The reviewer judges the paper favorably on soundness, significance, and originality, calls the experimental scope "very impressive" ($480$ runs across $16$ open-weight LLMs), identifies the decoupling of compression rate from bit-width as "a novel and interesting idea," highlights the data-free property as "practically useful," and recognizes the on-the-fly block-wise decompression as "a good contribution to systems research." The highlighted weaknesses are presentation issues only: moving implementation details and memory footprint reporting to the main paper, and expanding throughput evaluation to longer sequences and generation lengths. We address all of these below and will incorporate the changes in the camera-ready paper. We would appreciate it if the reviewer would consider raising their score in light of these responses.
>
> ### **W1/Q1: Decompression pipeline and temporary buffer size should be stated clearly in the main paper**
>
> We agree that the block-wise decompression strategy described in Section A.1 is an important practical detail. In the revision, we will move the key description, that all weights of a transformer block are jointly compressed into a single bitstream and decompressed into a reusable buffer before each block's forward pass, into the main text, alongside Algorithm 2\. We will also include a brief description of how tensor views avoid memory copies (Line 2 of Algorithm 2). We note that this pipeline follows the block-wise decompression approach suggested in the DFloat11 paper [1], and we will add a proper reference to it in the main text. The NVIDIA Nsight profiling visualization (Figure A.2) will be referenced from the main text to support our efficiency claims.
>
> ### **W2/Q2: Full inference-time memory footprint table should appear in the main paper**
>
> We agree. Due to space constraints in the main paper, the full memory analysis was moved to the appendix, where Figure F.3 already reports peak memory across all configurations. In the revision, we will add a condensed summary table in the main text reporting: (1) compressed weight storage, (2) scale parameter overhead, (3) ANS metadata, (4) decompression buffer size (one transformer block in Float8), (5) peak GPU memory during inference, and (6) KV cache. For LLaMA-2 $70$B at $2.1$ bits, the model weights occupy $\sim 18.8$ GiB, scales add $\ll 1\%$, ANS metadata is negligible, and the decompression buffer is $\sim 0.8$ GiB. We thank the reviewer for this concrete suggestion.
>
> ### **W3/Q3: Speed evaluation should cover longer context and generation lengths**
>
> We appreciate this point. Due to space constraints, the full throughput analysis was moved to the appendix, where Figures F.1–F.3 already include results for varying sequence lengths (up to $4096$ for $7$B models) and generation lengths (up to $256$ tokens). We will move a representative subset into the main text. We note that LLaMA-2's native context window is $4096$ tokens (not $8192$); however, we are happy evaluate at $8192$ tokens on models that natively support longer contexts (e.g., LLaMA-3). We would like to emphasize that decompression cost is independent of sequence length: weights are decompressed once per block per forward pass regardless of how many tokens are processed, so longer sequences amortize the overhead more effectively. The appendix figures confirm that EntQuant's relative throughput compared to baselines is stable or improves with longer contexts.
>
> ### **Missing societal impact discussion**
>
> We thank the reviewer for this thoughtful point. We agree that enabling stronger models on lower-resource hardware has dual-use implications that deserve explicit discussion. In the revision, we will expand the Impact Statement to more thoroughly address both the benefits and the potential risks of lowering the computational barrier to deploying powerful models.
>
> ### References
>
> \[1\] Zhang et al. 70% Size, 100% Accuracy: Lossless LLM Compression for Efficient GPU Inference via Dynamic-Length Float (DFloat11). NeurIPS, 2025.

---

> > ### Author Rebuttal · Reviewer_ZKAA · 2026-04-02
> >
> > Thank you to the authors for the rebuttal. My concerns have been addressed as follows:
> > (1) The proposed move of technical details from the appendix to the main paper resolves the clarity regarding representation issues.
> > (2) The addition of Llama 3 evaluations with 8192 tokens will provide necessary evidence of the method’s effectiveness in long-context scenarios.
> > Based on these responses, I believe the paper is stronger now. I support its acceptance.

---

> > > ### Author Response · Authors · 2026-04-07
> > >
> > > We thank the reviewer for the positive and constructive feedback. We are pleased that all raised concerns have been successfully addressed. As the reviewer notes the paper is now stronger and merits acceptance, we would be grateful if the score could be increased beyond a Weak Accept (4) to reflect this updated assessment. We appreciate the reviewer's time and support of our work.

---

### Official Review · Reviewer_c9S6 · 2026-03-10

**Soundness:** 3
**Presentation:** 3
**Significance:** 2
**Originality:** 3
**Overall Recommendation:** 3
**Confidence:** 4

**Summary:**

The paper presents a data-free LLM compression method: instead of treating entropy coding as a post hoc compression add-on, it explicitly optimizes quantized weights for low entropy and then decodes them on the fly during inference. That framing is interesting, and the results in the data-free extreme-compression regime are clearly nontrivial. In particular, the paper shows that EntQuant remains functional around effective 2-bit rates where simple data-free baselines collapse.

**Compliance With Llm Reviewing Policy:**

Affirmed.

**Final Justification:**

I find the method interesting overall. The introduction of entropy coding appears to be a natural design choice, especially from the perspective of signal processing and data compression, where such an idea would be fairly straightforward to consider. However, I do not think the paper discusses the cost of this design choice in sufficient depth. In particular, the practical overhead associated with entropy coding, including complexity and efficiency trade-offs, would benefit from a clearer and more thorough discussion.

**Key Questions For Authors:**

Q1. The paper evaluates throughput/latency mainly against weight-only data-free baselines such as NF4 and HQQ, but not against strong calibration-based PTQ methods such as Quip#. Since these are widely used practical baselines, could the authors provide a direct comparison in serving-time metrics (e.g., prefill latency, decode latency, and throughput) to clarify the real deployment-time trade-off?

Q2. A major advantage of EntQuant is that it avoids calibration data and achieves very fast offline compression. However, calibration is often a one-time preprocessing cost, whereas serving-time latency/throughput affects every downstream user query. Could the authors clarify the target deployment scenarios in which a fully data-free method is especially necessary or clearly preferable to calibration-based PTQ? For example, are there practical settings where calibration data is unavailable, legally restricted, distribution-mismatched, or unsafe to use, such that EntQuant becomes uniquely suitable?

It would strengthen the paper if the authors could explicitly discuss in the main text when a fully data-free method is truly necessary or especially preferable. Clarifying these target scenarios would make the practical value of the proposed method much easier for readers to understand.

My main concern is that model compression is often valued not only for reducing memory/storage cost, but also for improving serving-time efficiency. While EntQuant is compelling in terms of data-free compression and memory reduction, it is less clear whether it can provide serving-time efficiency gains comparable to current low-bit PTQ methods, especially given the added on-the-fly decoding overhead. This may limit the applicability and practical value of the method. If the authors can convincingly address this concern, I would be happy to raise my score.

**Limitations:**

Yes

**Strengths And Weaknesses:**

**Strength**

1) The idea of decoupling compression rate from quantization bit-width is interesting. By explicitly regularizing quantized weights toward low entropy and then entropy-coding them, the method avoids the rigidity of fixed low-bit representations while retaining higher representational expressiveness.

2) The method is data-free and requires neither calibration data nor retraining. The reported compression time of less than 30 minutes for a 70B model is substantially lower than the hours-to-tens-of-hours cost of several calibration- or training-based baselines, while maintaining competitive accuracy.

3) The experimental results in the extreme compression regime are particularly strong. At around 2-bit effective rates, EntQuant clearly outperforms prior data-free baselines such as HQQ, which tend to suffer from functional collapse, showing that the proposed entropy-aware formulation is especially effective in this challenging setting.

**Weakness**

1) The efficiency evaluation is incomplete. While the paper reports throughput in the main text and latency in the appendix, these measurements are only compared against BFloat16/Float8/NF4/HQQ-style baselines, not against strong calibration-based PTQ methods such as Quip#. As a result, the practical serving-time trade-off relative to mainstream PTQ baselines remains unclear.

2) The claimed runtime advantage mainly concerns offline compression, which is a one-time cost during model preparation rather than a recurring cost during deployment. For many practical users, serving-time latency and throughput may matter more than compression runtime.

3) The method is more compelling as a storage/compression approach than as an inference-acceleration method. The default setup is weight-only quantization, and on-the-fly entropy decoding introduces measurable overhead; the paper reports EntQuant to be about 1.5–2× slower than BFloat16.

---

> ### Author Rebuttal · Authors · 2026-03-27
>
> We thank Reviewer c9S6 for the constructive review. The reviewer acknowledges the soundness, presentation, and originality of the work, finds the decoupling idea "interesting," notes that our $<30$-minute data-free compression is "substantially lower" than calibration-based alternatives, and calls the extreme-compression results "clearly nontrivial," with EntQuant "clearly outperform\[ing\] prior data-free baselines." The concerns center on serving-time efficiency and deployment scenario justification. We address these below.
>
> ### **W1/Q1: No serving-time comparison with calibration-based PTQ methods like QuIP\#**
>
> We appreciate this point. A direct comparison is difficult because the methods use fundamentally different inference kernels (QuIP\#: lattice codebook lookups; EntQuant: ANS decoding + Float8 Marlin), and fair numbers require integrating both into a unified serving framework like vLLM. We provide comprehensive standalone throughput and latency across multiple batch sizes, sequence lengths, and generation lengths (Figures F.1–F.3). The key comparison is the **full trade-off**: EntQuant achieves comparable accuracy (Table D.1) with zero data and $<30$ min compression, vs. $\sim 50$ hours on $8\times$ A100s for QuIP\#.
>
> ### **W2: Compression speed is a one-time cost, not a deployment advantage**
>
> We do not claim an inference-time advantage. Our claims are: (1) data-free extreme compression with preserved quality, and (2) practical deployment with acceptable overhead. Compression speed enables a fundamentally different workflow: immediate compression and testing of new models without data collection or multi-day calibration, a significant recurring advantage given weekly model releases (see also Q2 below).
>
> ### **W3: EntQuant is more compelling as storage/compression than inference-acceleration**
>
> We agree EntQuant is primarily a memory compression method. However, memory compression **enables inference**: a model that does not fit in VRAM cannot run at all, or must use CPU offloading ($3 - 45\times$ slower). The practical comparison is often "compressed $70$B model vs. uncompressed $13$B model" or "EntQuant on GPU vs. BFloat16 with CPU offloading." Fewer GPUs also directly reduces hardware costs.
>
> DFloat11 [4], a recognized lossless LLM compression method, operates under the same on-the-fly decompression paradigm with similar overhead, suggesting this trade-off is broadly accepted. Our throughput already matches NF4 (Figure 5). As discussed in our response to Reviewer nGS2 (W3), optimized kernels typically follow algorithmic contributions, and fused decompression-GEMM kernels are a natural next step.
>
> ### **Q2: Target deployment scenarios where a data-free method is necessary**
>
> We thank the reviewer for this very useful suggestion, which directly helps highlight EntQuant's strengths. We confirm that the practical settings the reviewer outlines all arise in practice. We identify four concrete deployment regimes where EntQuant is especially valuable:
>
> **(1) Resource-constrained deployment:** In self-hosting scenarios, memory is typically the binding constraint. EntQuant enables running a $70$B model on a single $32$ GiB consumer GPU at $3$ bits, where the alternative would be a smaller model, CPU offloading ($3 - 45\times$ slower, Section 3.4), or multi-GPU serving. Compression also reduces GPU requirements and hardware costs.
>
> **(2) Unavailable or restricted calibration data:** Models like LLaMA-3.3 $70$B Instruct or Mistral Large are fine-tuned on proprietary data users cannot access. Similarly, in regulated domains (healthcare, finance), data protection regulations such as GDPR's purpose limitation principle (Art. 5(1)(b)) or industry-specific compliance rules may prohibit repurposing training data for calibration. EntQuant compresses directly from weights alone.
>
> **(3) Safety-tuned and reasoning models:** Models trained with RLHF or constitutional AI have delicate internal representations; calibration-based quantization can degrade safety-aligned behaviors unpredictably [1, 2, 3]. EntQuant sidesteps this risk entirely.
>
> **(4) Rapid model iteration:** In the current open-weight ecosystem, where $1000$+ models are uploaded to Hugging Face daily, hours-long calibration pipelines create recurring bottlenecks that a $<30$-minute data-free method eliminates. This also covers domain-adapted models where proxy calibration data drawn from a different distribution risks degrading the very capabilities the model was fine-tuned to acquire.
>
> We will add a dedicated paragraph in the main text and agree with the reviewer that making these use cases explicit will substantially clarify EntQuant's practical value.
>
> ### References
>
> \[1\] Lee et al. Quantization Trade-Offs in LLMs. IJCAI, 2025.
>
> \[2\] Wee et al. Alignment-Aware Quantization. arXiv:2511.07842, 2025.
>
> \[3\] Kharinaev et al. Quantization Impact on LLM Safety. arXiv:2502.15799, 2025.
>
> \[4\] Zhang et al. DFloat11: Lossless LLM Compression. NeurIPS, 2025.

---

> > ### Author Rebuttal · Reviewer_c9S6 · 2026-04-02
> >
> > Thank you for the rebuttal. I agree that this is an interesting weight-only compression method that can enable deployment of larger models on GPUs with limited memory, but this also narrows the scope of its practical benefit. My main concern remains that the paper does not provide the key missing evidence: a direct latency comparison against strong practical PTQ baselines at comparable quality/memory operating points. While such baselines require calibration, calibration is typically a one-time preprocessing cost and is still much cheaper than QAT, even if it is more expensive than fully data-free methods. Once quantized, the resulting weights can be distributed and reused without repeating that cost. The rebuttal helps clarify the paper’s scope, but it mainly reframes the contribution around memory savings, offline compression speed, and data-free deployment, rather than resolving the missing serving-time trade-off. For this reason, while I appreciate the novelty and promise of the work, I am maintaining my original weak reject score.

---

> > > ### Author Response · Authors · 2026-04-08
> > >
> > > We sincerely thank the reviewer for the continued engagement. Before addressing the serving-time concern, we would like to highlight that EntQuant is **state-of-the-art in the data-free regime**, matching QuIP\# and AQLM in accuracy (Table D.1).
> > >
> > > ### Serving-time trade-off: two distinct deployment scenarios
> > >
> > > The reviewer is right that serving-time efficiency matters. We acknowledge that Table 3(a) in the paper focused on compression time and did not include inference-time comparisons against strong quantization baselines. How impactful the inference slowdown is, however, depends on the deployment scenario. To have a clear basis for this conversation, it is important to distinguish two scenarios, which we will also make explicit in the revision:
> > >
> > > **Scenario A: Large-scale multi-user serving.** Per-token latency is critical and optimized kernels (e.g., Marlin for W4A16) provide clear advantages. We acknowledge that EntQuant is not optimized for this regime, and we will make this limitation explicit in the revision.
> > >
> > > **Scenario B: Resource-constrained self-hosting.** Users in this regime face strict memory limitations and typically lack the hardware, data, or expertise to run calibration-based or fine-tuning-based compression themselves. EntQuant enables these users to both compress and deploy, e.g., a $70$B model on a single consumer GPU without any data or multi-GPU infrastructure. The data-free, safety, and privacy advantages (detailed in our initial response Q2) further strengthen the case.
> > >
> > > ### Proposed serving-time comparison table
> > >
> > > We commit to adding a serving-time comparison table in the camera-ready. Below is a preliminary version:
> > >
> > > | Method | Bits | TPOT $7/8$B | Hardware |
> > > | :--- | :---: | :---: | :--- |
> > > | GPTQ [5] | $4$ | $\sim 0.6\times$ | H200 |
> > > | GPTQ+Marlin [2] | $4$ | $\sim 2.9\times$ | A10 |
> > > | QuIP [6] | $2$ | $\sim 0.4\times$* | A6000 |
> > > | QuIP\# [3] | $2$ | $\sim 3.2\times$ | RTX 4090 |
> > > | AQLM [4] | $2$ | $\sim 0.6\times$ | RTX 4090 |
> > > | DFloat11 [1] | $\sim 11$ | $\sim 0.6\times$ | A100 |
> > > | EntQuant | $2$ | $\sim 0.6\times$ | H100 |
> > >
> > > TPOT (time per output token) measures decode latency; TTFT (time to first token) measures prefill latency. This table is abbreviated for the rebuttal, reporting only TPOT ratios (quantized $\div$ uncompressed, $>1\times$ = faster) at batch size $1$ for $7/8$B models. The camera-ready will include the full table with $70$B results and TTFT ratios. *Estimated from [6] reporting $1.5\times$ slower than GPTQ on OPT-66B. All other ratios are grounded in published results: GPTQ from JarvisLabs vLLM benchmarks [7]; QuIP from [6]; QuIP\#/AQLM from Table 6 of [3]; DFloat11 from Figure 10 of [1].
> > >
> > > Key observations:
> > >
> > > (1) **Historical pattern: kernel optimization closes the gap.** GPTQ without Marlin ($\sim 0.6\times$) and QuIP without E8P ($\sim 0.4\times$) were both slower than uncompressed inference. Optimized kernels (Marlin, E8P) later transformed these into speedups ($\sim 2.9\times$ and $\sim 3.2\times$). EntQuant is at the same pre-kernel-optimization stage.
> > >
> > > (2) **EntQuant's overhead matches established methods.** AQLM, GPTQ without Marlin, and QuIP all show similar $\sim 0.4$-$0.6\times$ TPOT, indicating that the overhead is not an outlier due to entropy coding.
> > >
> > > (3) **EntQuant scales better than DFloat11.** Both use on-the-fly decompression and show $\sim 0.6\times$ at $7/8$B, but at $70$B DFloat11 drops to $\sim 0.3\times$ on $4\times$ A100 while EntQuant fits on a single GPU at $\sim 18.8$ GiB.
> > >
> > > In their original assessment, the reviewer generously stated: *"If the authors can convincingly address this concern, I would be happy to raise my score."* We hope the evidence above addresses this: we provide (a) state-of-the-art accuracy matching calibration-based methods (Table D.1), (b) an explicit distinction between deployment scenarios, (c) a concrete serving-time comparison grounded in published numbers showing EntQuant is competitive. Additionally, if the reviewer would find it helpful, we are happy to benchmark all methods from the table under a unified single-GPU H100 setting at batch size $1$ to provide a fully controlled comparison. Together with the support from the other three reviewers, we respectfully ask the reviewer to reconsider their score.
> > >
> > > ### References
> > >
> > > \[1\] Zhang et al. DFloat11: Lossless LLM Compression. NeurIPS, 2025.
> > >
> > > \[2\] Frantar et al. MARLIN: Mixed-Precision Auto-Regressive Parallel Inference. PPoPP, 2025.
> > >
> > > \[3\] Tseng et al. QuIP\#: Even Better LLM Quantization with Hadamard Incoherence and Lattice Codebooks. ICML, 2024.
> > >
> > > \[4\] Egiazarian et al. Extreme Compression of Large Language Models via Additive Quantization. ICML, 2024.
> > >
> > > \[5\] Frantar et al. GPTQ: Accurate Post-Training Quantization for Generative Pre-Trained Transformers. ICLR, 2023.
> > >
> > > \[6\] Chee et al. QuIP: 2-Bit Quantization of Large Language Models With Guarantees. NeurIPS, 2023.
> > >
> > > \[7\] JarvisLabs. vLLM Quantization: Complete Guide with Benchmarks.

---

### Official Review · Reviewer_RLVb · 2026-03-12

**Soundness:** 3
**Presentation:** 3
**Significance:** 3
**Originality:** 3
**Overall Recommendation:** 4
**Confidence:** 4

**Summary:**

The paper proposes EntQuant, a data-free post-training compression framework that decouples numerical precision from storage cost through entropy coding. Instead of directly quantizing weights to extremely low precision, the method keeps an 8-bit base representation, optimizes channel-wise scales with a reconstruction-plus-regularization objective to induce low-entropy quantized weights, and then applies ANS entropy coding to achieve near-2-bit effective storage. Experiments on both base and instruction-tuned LLMs show that the method substantially improves the compression–accuracy trade-off in the extreme low-bit regime, while remaining calibration-free and relatively efficient to apply.

**Compliance With Llm Reviewing Policy:**

Affirmed.

**Key Questions For Authors:**

1.Can you provide more direct evidence that entropy reduction itself, rather than retained byte-level expressivity or induced sparsity, is the main driver of the gains?
2.Can you provide layer-wise analysis connecting entropy, reconstruction error, and downstream accuracy?
3.In what deployment regimes is EntQuant preferable to standard low-bit quantization given the decoding overhead?

**Limitations:**

yes

**Strengths And Weaknesses:**

Strengths:
The paper addresses an important and practical problem: enabling extreme model compression without calibration data or recovery training. This is a meaningful setting where many existing data-free PTQ methods degrade sharply.

The central idea of decoupling storage rate from numerical precision via entropy coding is conceptually clean and practically relevant. The method provides a useful alternative perspective to standard low-bit quantization, which directly ties compression to representational precision.

The approach is simple and deployable. It only optimizes scale parameters rather than requiring full retraining, and the paper includes a concrete inference-time design with on-device decoding rather than stopping at offline compression.

The empirical evaluation is fairly broad and includes comparisons against both data-free and data-dependent baselines, as well as results on instruction-tuned models. The reported gains in the 2–3 bit regime are particularly strong and appear practically meaningful.

Weaknesses:
The practical motivation is clear: existing data-free PTQ methods degrade sharply under extreme bit budgets, while data-dependent methods incur substantial calibration or retraining cost. However, the method-level justification is still incomplete. Although the paper provides some conceptual and empirical support for entropy minimization, it remains unclear whether the gains primarily come from lower entropy itself, from induced sparsity, or from retaining a byte-level representation with higher expressivity than fixed low-bit quantization. More diagnostic analysis would strengthen this claim.

It would be stronger with more diagnostic analysis. For example, layer-wise evidence linking entropy reduction, reconstruction error, and downstream accuracy would help clarify whether entropy-aware quantization is the primary driver of the reported gains.

Although the method achieves strong memory savings, it introduces inference-time decoding overhead. The paper discusses this trade-off to some extent, but it does not yet fully characterize the deployment settings in which this design is preferable to standard quantization approaches.

---

> ### Author Rebuttal · Authors · 2026-03-27
>
> We thank Reviewer RLVb for the detailed and positive review. The reviewer gives a strong assessment across all four criteria (soundness, presentation, significance, and originality), describes the problem as "important and practical," the central idea as "conceptually clean and practically relevant," the approach as "simple and deployable," and the empirical gains in the $2 - 3$ bit regime as "particularly strong and appear practically meaningful." The weaknesses concern diagnostic evidence: distinguishing whether gains arise from entropy reduction, induced sparsity, or retained byte-level expressivity; layer-wise analysis linking entropy and accuracy; and a clearer characterization of deployment regimes where EntQuant is preferable to standard low-bit quantization. Below, we address the questions of the reviewer, which are aligned with the weaknesses mentioned above.
>
> ### **Q1: Is the gain primarily from lower entropy, induced sparsity, or retained byte-level expressivity?**
>
> We appreciate this question and acknowledge that fully disentangling these factors experimentally is challenging, as they are by design intertwined in our method. However, we can provide a quantitative argument that **sparsity alone cannot account for the observed entropy reductions**.
>
> Consider two sparsity-only baselines where a fraction $s$ of weights are zeroed: (1) non-zero values remain uniform over $255$ Float8 levels, giving $H(s) = -s \cdot \log_2(s) + (1 - s) \cdot \log_2(255)$; (2) non-zero values keep their original distribution at $\sim 6.5$ bits (Figure 4), giving $H(s) = -s \cdot \log_2(s) + (1 - s) \cdot 6.5$.
>
> | | $s{=}0.1$ | $s{=}0.2$ | $s{=}0.3$ | $s{=}0.4$ | $s{=}0.5$ | $s{=}0.6$ | $s{=}0.7$ | $s{=}0.8$ |
> | :--- | :---: | :---: | :---: | :---: | :---: | :---: | :---: | :---: |
> | Sparsity-with-uniform $H$ (bits) | $7.53$ | $6.86$ | $6.12$ | $5.33$ | $4.50$ | $3.64$ | $2.76$ | $1.86$ |
> | Sparsity-unchanged-distribution $H$ (bits) | $6.18$ | $5.66$ | $5.07$ | $4.43$ | $3.75$ | $3.04$ | $2.31$ | $1.56$ |
>
> Comparing these theoretical values to Figure B.1, it is apparent that EntQuant achieves entropies **substantially lower than what sparsity alone would predict** at corresponding sparsity levels. To achieve $2$ bits, both scenarios would require sparsity of more than $0.7$. EntQuant, on the other hand, achieves $2$ bits at sparsity of only $0.4$. This demonstrates that $\ell_1$ regularization **reshapes the non-zero weight distribution** toward lower entropy beyond the sparsity effect. Table 1 further shows that at $2$ bits, EntQuant retains $\sim 35$ unique values vs. only $4$ for fixed $2$-bit quantization, preserving the expressivity that prevents functional collapse (Table 2). We will add this analysis to the appendix.
>
> ### **Q2: Layer-wise analysis linking entropy, reconstruction error, and downstream accuracy**
>
> We thank the reviewer for this suggestion. Figure A.1 already demonstrates that the mapping from $\lambda$ to entropy is remarkably consistent across layers and models. We will supplement this with a per-layer breakdown of entropy achieved and reconstruction error. We note that correlating individual layer entropy with downstream accuracy is methodologically challenging, as layers interact non-linearly during inference, making it difficult to attribute accuracy changes to a single layer's entropy in isolation. We are happy to discuss the best way to present this analysis with the reviewer.
>
> ### **Q3: In what deployment regimes is EntQuant preferable?**
>
> We believe that EntQuant is most valuable in the scenarios discussed in detail in our response to Reviewer c9S6 (Q2), where we identify four concrete deployment regimes. The two most relevant to this reviewer's question are:
>
> **(1) Resource-constrained deployment:** EntQuant enables running a $70$B model on a single $32$ GiB consumer GPU at $3$ bits, where the alternative is a smaller model, CPU offloading ($3 - 45\times$ slower), or multi-GPU serving.
>
> **(2) Rapid model iteration:** In the current open-weight ecosystem with weekly frontier releases, hours-long calibration pipelines create recurring bottlenecks that EntQuant's $<30$-minute data-free compression eliminates.
>
> We will add a dedicated paragraph outlining these scenarios in the main text.

---

> > ### Author Rebuttal · Reviewer_RLVb · 2026-04-03
> >
> > I have read the rebuttal. I will keep my rate unchanged.

---

> > > ### Author Response · Authors · 2026-04-08
> > >
> > > We thank the reviewer for their time and for reading our rebuttal. We are glad that our response partially resolved their concerns.
> > >
> > > We would have liked to address the remaining issues, but from the current feedback it is not clear to us which specific concerns remain open. In our rebuttal, we provided:
> > >
> > > - (a) A quantitative argument that sparsity alone cannot account for the observed entropy reductions, which we will add to the appendix (Q1).
> > > - (b) A commitment to add a per-layer breakdown of entropy and reconstruction error (Q2).
> > > - (c) Concrete deployment scenarios where EntQuant is preferable, which we will discuss in the main text (Q3, also elaborated in our response to Reviewer c9S6).
> > >
> > > We hope the reviewer finds these responses sufficient, and we remain committed to incorporating the promised analyses in the revision.

---

### Official Review · Reviewer_nGS2 · 2026-03-12

**Soundness:** 3
**Presentation:** 3
**Significance:** 3
**Originality:** 3
**Overall Recommendation:** 4
**Confidence:** 3

**Summary:**

The paper introduces EntQuant which is a post training quantization method designed to decouple the compression rate from the representational bit width. Instead of forcing weights into strict low bit formats to save space the method keeps weights in higher precision Float8 or Int8 formats but optimizes their channel wise scales to minimize entropy. The l1 norm is utilized as a differentiable proxy for entropy during this optimization step. The optimized weights are then compressed using an Asymmetric Numeral Systems coder. During inference the weights are decompressed on the fly directly in the GPU VRAM. The authors show that this data free approach achieves an effective 2 bit compression rate without the severe functional collapse usually seen in other data free methods like HQQ or NF4. It also effectively preserves the capabilities of instruction tuned models.

**Compliance With Llm Reviewing Policy:**

Affirmed.

**Key Questions For Authors:**

1. Could you elaborate on whether alternative differentiable proxies for entropy were tested and why the l1 norm was ultimately selected beyond its robustness to outliers

2. How does the on the fly decompression scale with sequence length during the prefill phase and are there specific context length limits where the block wise decoding buffer becomes a strict bottleneck

**Limitations:**

The authors adequately discuss the limitations of their work. They explicitly mention the intentional simplicity of their method and clearly acknowledge the latency overhead introduced by the on the fly decompression step. They also openly note that their evaluation budget restricted testing to a certain scale of models and tasks requiring further validation on larger mixture of experts architectures.

**Strengths And Weaknesses:**

Strengths
- The paper proposes to decouple storage bit rate from arithmetic precision by optimizing entropy during quantization, which provides an interesting perspective on model compression.
- The method operates entirely data free which is an advantage for compressing instruction tuned or specialized reasoning models where using calibration data might ruin specific learned capabilities.
- The empirical results are convincing especially regarding the preservation of performance at the 2 bit boundary where standard rounding or half quadratic quantization methods completely break down.

Weaknesses
- The evaluation lacks comparisons with recent vector quantization methods for LLMs, such as VQ-LLM and CommVQ. Benchmarking against these newer baselines would better demonstrate the method's actual competitiveness in extreme low-bit compression.
- The optimization objective relies heavily on the l1 norm as a surrogate for entropy which the authors justify mainly empirically and with a basic worst case bound. A deeper theoretical link showing exactly why l1 specifically induces the right kind of clustering for ANS encoding would strengthen the work.
- The inference speed is inevitably slower than standard fixed bit width quantization dropping throughput by a factor of 1.5 to 2 compared to uncompressed BFloat16. While the authors argue memory is the main bottleneck for self hosting users this latency penalty might limit adoption in high throughput serving scenarios.

---

> ### Author Rebuttal · Authors · 2026-03-27
>
> We thank Reviewer nGS2 for the careful reading and positive assessment. The reviewer views the paper favorably in all four evaluation dimensions (soundness, presentation, significance, and originality) and finds the decoupling of storage bit rate from arithmetic precision "an interesting perspective," notes the data-free property as "an advantage for compressing instruction tuned or specialized reasoning models," and finds the empirical results "convincing especially regarding the preservation of performance at the $2$-bit boundary." The concerns are: comparisons with VQ-LLM and CommVQ, a deeper theoretical link between the $\ell_1$ norm and entropy encoding, and the $1.5 - 2\times$ throughput overhead. We address them below.
>
> ### **W1: Lack of comparisons with recent vector quantization methods (VQ-LLM, CommVQ)**
>
> We thank the reviewer for raising this point, but we believe there is an important distinction to clarify: **neither VQ-LLM nor CommVQ proposes a weight compression algorithm**. VQ-LLM [1] is a systems contribution providing optimized inference kernels for existing VQ methods; it does not introduce a new compression technique. CommVQ [2] targets **KV-cache compression**, a complementary problem to weight compression.
>
> For weight-compression VQ baselines, our paper already compares against AQLM and QuIP\# (Table D.1), the leading methods. EntQuant matches their accuracy (Table 3) without any data requirements. We will add a clarifying note in the related work to distinguish weight compression from KV-cache compression.
>
> ### **W2/Q1: The $\ell_1$-norm as entropy proxy needs deeper theoretical justification**
>
> We direct the reviewer's attention to Section B.2, where we derive a formal worst-case bound via the maximum entropy principle: $H(W_q) \leq \lambda \mathbb{E}[|W_q|_1] + MN \cdot \log Z(\lambda)$. This bound is sharp and attained by distributions with density $q(x) = Z \exp(-\lambda |x|)$, the discrete analog of the standard Laplace assumption in quantization [3]. Standard $p$-norm inequalities yield analogous (but less sharp) bounds for other regularizers.
>
> The practical motivation is that LLM weight distributions are **unimodal**, making a sparsity-inducing prior a natural fit. We also experimented with an **MLP-based learned entropy proxy**, which added complexity without improving over $\ell_1$. A general optimality comparison across regularizer classes would require additional distributional assumptions and constitutes an interesting direction for future theoretical work. We will expand this discussion in the revision.
>
> ### **W3: Inference speed is $1.5 - 2\times$ slower than BFloat16**
>
> We acknowledge this overhead, but wish to draw an important historical parallel. **Innovation in model compression often advances along one efficiency axis at a time.** A concrete example is the Marlin kernel [4]: before its introduction, $\text{W4A16}$ quantization (AWQ, bitsandbytes) was even slower than uncompressed $\text{W16A16}$ inference. The community recognized the compression/accuracy gains, and optimized kernels followed. EntQuant is at an analogous stage: we have established a new compression/accuracy frontier, and fused decompression-GEMM kernels are a natural next step. Our throughput already matches NF4 (Figure 5), which is a very common consumer standard. The same on-the-fly decompression paradigm is used by DFloat11 with similar overhead (see our response to Reviewer c9S6, W3), suggesting this trade-off is broadly accepted.
>
> For the primary target of single-user self-hosting, memory is the binding constraint. EntQuant enables running a $70$B model on a $32$ GiB consumer GPU, where the alternative is CPU offloading at $3 - 45\times$ slower speeds (Section 3.4). We discuss additional deployment scenarios in our response to Reviewer c9S6 (Q2). We are happy to expand the discussion of inference overhead in Section 5.
>
> ### **Q2: How does decompression scale with sequence length during prefill?**
>
> The decompression cost is **independent of sequence length**. We decompress all weights of a transformer block once before its forward pass, then reuse them for all tokens (Section A.1). The buffer is fixed-size (one transformer block), so longer sequences amortize the overhead more effectively. Figures F.1–F.3 confirm that throughput scales with batch and sequence length identically to baseline methods.
>
> ### References
>
> \[1\] Liu et al. VQ-LLM: Code Generation for VQ-Augmented LLM Inference. HPCA, 2025.
>
> \[2\] Li et al. CommVQ: Commutative VQ for KV Cache Compression. ICML, 2025.
>
> \[3\] Banner et al. Post Training 4-bit Quantization of CNNs. NeurIPS, 2019.
>
> \[4\] Frantar et al. MARLIN: Mixed-Precision Auto-Regressive Parallel Inference. PPoPP, 2025.

---

> > ### Author Rebuttal · Reviewer_nGS2 · 2026-04-07
> >
> > Thank authors for the effort on their rebuttal. My question is partially resolved. I decide to keep my score.

---

> > > ### Author Response · Authors · 2026-04-08
> > >
> > > We thank the reviewer for acknowledging our rebuttal and for their time throughout the review process. We are glad that our response partially resolved their concerns.
> > >
> > > We would have liked to address the remaining concerns, but from the current feedback it is not entirely clear to us which specific issues remain open. We hope that our original rebuttal, together with our response to Reviewer c9S6 (which provides a detailed serving-time comparison and discusses two distinct deployment scenarios), addresses the reviewer's concerns about inference overhead.

---

### Decision · Program_Chairs · 2026-04-30

**Decision:**

Accept (regular)

**Comment:**

This paper provides a post-training copression method which quantizes weights with different bitrates. To do so it first quantizes into an 8-bit representation for which it finds appropriate quantization scales by solving an optimization problem, after which it uses entropy coding to achieve nearly 2-bits storage/weight on average.

Reviewers agreed that this idea is new and interesting, and found that the empirical results are strong especially in extreme compression regimes, where other data-free methods (which do not require some prior calibration) generally fail. The experimental effort (480 runs across 16 LLMs) was commended.

The main concern was serving-time efficiency, as decoding introduces overhead. In this context, the issue that was raised was that comparisons against calibration-based PTQ methods is lacking. The authors contended that this method's scope is mostly relevant to resource-constrained scenarios (as well as to the case where compression needs to be performed across changing models, and where calibration is prohibitive).

Overall, the contribution was found meaningful, and the experiments well developed. Despite the concerns regarding serving-time efficiency, the general conclusion is that this paper has clear and well supported results.